# Learning Policies with Zero or Bounded Constraint Violation for Constrained MDPs

**Tao Liu**[*]
Texas A&M University
tliu@tamu.edu

**Ruida Zhou**[*]
Texas A&M University
ruida@tamu.edu

**Dileep Kalathil**
Texas A&M University
dileep.kalathil@tamu.edu

**P. R. Kumar**
Texas A&M University
prk@tamu.edu

**Chao Tian**
Texas A&M University
chao.tian@tamu.edu

## Abstract

We address the issue of safety in reinforcement learning. We pose the problem in an episodic framework of a constrained Markov decision process. Existing results have shown that it is possible to achieve a reward regret of $\tilde{\mathcal{O}}(\sqrt{K})$ while allowing an $\tilde{\mathcal{O}}(\sqrt{K})$ constraint violation in $K$ episodes. A critical question that arises is whether it is possible to keep the constraint violation even smaller. We show that when a strictly safe policy is known, then one can confine the system to zero constraint violation with arbitrarily high probability while keeping the reward regret of order $\tilde{\mathcal{O}}(\sqrt{K})$. The algorithm which does so employs the principle of optimistic pessimism in the face of uncertainty to achieve safe exploration. When no strictly safe policy is known, though one is known to exist, then it is possible to restrict the system to bounded constraint violation with arbitrarily high probability. This is shown to be realized by a primal-dual algorithm with an optimistic primal estimate and a pessimistic dual update.

## 1 Introduction

Reinforcement learning (RL) addresses the problem of learning an optimal control policy that maximizes the expected cumulative reward while interacting with an unknown environment [25]. Standard RL algorithms typically focus only on maximizing a single objective. However, in many real-world applications, the control policy learned by an RL algorithm has to additionally satisfy stringent safety constraints [9, 4]. For example, an autonomous vehicle may need to reach its destination in the minimum possible time without violating safety constraints such as crossing the middle of the road. The Constrained Markov Decision Process (CMDP) [2, 12] formalism, where one seeks to maximize a reward while satisfying safety constraints, is a standard approach for modeling the necessary safety criteria of a control problem via constraints on cumulative costs.

Several policy-gradient-based algorithms have been proposed to solve CMDPs. Lagrangian-based methods [26, 24, 21, 16] formulate the CMDP problem as a saddle-point problem and optimize it via primal-dual methods, while Constrained Policy Optimization [1, 29] (inspired by the trust region policy optimization [23]) computes new dual variables from scratch at each update to maintain constraints during learning. Although these algorithms provide ways to learn an optimal policy, performance guarantees about reward regret, safety violation or sample complexity are rare.

One class of RL algorithms for which performance guarantees are available follow the principle of Optimism in the Face of Uncertainty (OFU) [8, 10, 22], and provide an $\tilde{\mathcal{O}}(\sqrt{K})$ guarantee

---

[*]The first two authors contributed equally.

35th Conference on Neural Information Processing Systems (NeurIPS 2021).

Table 1: Regret and constraint violation comparisons for algorithms on episodic CMDPs

| Algorithm | Regret[2] | Constraint violation [2] |
|---|---|---|
| OPDOP [8] | $\tilde{\mathcal{O}}(H^3\sqrt{|\mathcal{S}|^2|\mathcal{A}|K})$ | $\tilde{\mathcal{O}}(H^3\sqrt{|\mathcal{S}|^2|\mathcal{A}|K})$ |
| OptCMDP [10] [3] | $\tilde{\mathcal{O}}(H^2\sqrt{|\mathcal{S}|^3|\mathcal{A}|K})$ | $\tilde{\mathcal{O}}(H^2\sqrt{|\mathcal{S}|^3|\mathcal{A}|K})$ |
| OptCMDP-bonus [10] [3] | $\tilde{\mathcal{O}}(H^2\sqrt{|\mathcal{S}|^3|\mathcal{A}|K})$ | $\tilde{\mathcal{O}}(H^2\sqrt{|\mathcal{S}|^3|\mathcal{A}|K})$ |
| OptDual-CMDP [10] [3] | $\tilde{\mathcal{O}}(H^2\sqrt{|\mathcal{S}|^3|\mathcal{A}|K})$ | $\tilde{\mathcal{O}}(H^2\sqrt{|\mathcal{S}|^3|\mathcal{A}|K})$ |
| OptPrimalDual-CMDP [10] [3] | $\tilde{\mathcal{O}}(H^2\sqrt{|\mathcal{S}|^3|\mathcal{A}|K})$ | $\tilde{\mathcal{O}}(H^2\sqrt{|\mathcal{S}|^3|\mathcal{A}|K})$ |
| C-UCRL [30] [4] | $\tilde{\mathcal{O}}(T^{\frac{3}{4}})$ | 0 |
| **OptPess-LP** | $\tilde{\mathcal{O}}(\frac{H^3}{\tau-c^0}\sqrt{|\mathcal{S}|^3|\mathcal{A}|K})$ | 0 |
| **OptPess-PrimalDual** | $\tilde{\mathcal{O}}(\frac{H^3}{\tau-c^0}\sqrt{|\mathcal{S}|^3|\mathcal{A}|K})$ | $\mathcal{O}(1)$ [5] |

for the reward regret, where $K$ is the number of episodes. However, these algorithms also have $\tilde{\mathcal{O}}(\sqrt{K})$ safety violations. Such significant violation of the safety constraints during learning may be unacceptable in many safety-critical real-world applications such as the control of autonomous vehicles or power systems. These applications demand a class of safe RL algorithms that can provably guarantee safety during learning. With this goal in mind, we aim to answer the following open theoretical question in this paper:

**Can we design safe RL algorithms that can achieve an $\tilde{\mathcal{O}}(\sqrt{K})$ regret with respect to the performance objective, while guaranteeing zero or bounded safety constraint violation with arbitrarily high probability?**

We answer the above question affirmatively by proposing two algorithms and establishing their stringent safety performance during learning. Our focus is on the tabular episodic constrained RL setting (unknown transition probabilities, rewards, and costs). The key idea behind both algorithms is a concept used earlier for safe exploration in constrained bandits [20, 17], which we call "Optimistic Pessimism in the Face of Uncertainty (OPFU)" here. The optimistic aspect incentivizes the algorithm for using exploration policies that can visit new state-action pairs, while the pessimistic aspect disincentivizes the algorithm from using exploration policies that can violate safety constraints. By carefully balancing optimism and pessimism, the proposed algorithms guarantee zero or bounded safety constraint violation during learning while achieving an $\tilde{\mathcal{O}}(\sqrt{K})$ regret with respect to the reward objective.

The two algorithms address two different classes of the safe learning problem: whether a strictly safe policy is known a priori or not. The resulting exploration strategies are very different in the two cases.

1. **OptPess-LP Algorithm:** This algorithm assumes the prior knowledge of a strictly safe policy. It ensures zero safety constraint violation during learning with high probability and utilizes the linear programming (LP) approach for solving a CMDP problem. The algorithm achieves a reward regret of $\tilde{\mathcal{O}}(\frac{H^3}{\tau-c^0}\sqrt{|\mathcal{S}|^3|\mathcal{A}|K})$ with respect to the performance objective, where $H$ is the number of steps per episode, $\tau$ is the given constraint on safety violation, $c^0$ is the known safety constraint value of a strictly safe policy $\pi^0$, and $|\mathcal{S}|$ and $|\mathcal{A}|$ are the number of states and actions respectively.

2. **OptPess-PrimalDual Algorithm:** This algorithm addresses the case where no strictly safe policy, but a feasible strictly safe cost is known. By allowing a bounded (in $K$) safety cost, it opens up space for exploration. The OptPess-PrimalDual algorithm avoids linear programming and its attendant complexity and exploits the primal-dual approach for solving a CMDP problem. The proposed approach improves the computational tractability, while

---

[2]This table is presented for $K \geq \text{poly}(|\mathcal{S}|, |\mathcal{A}|, H)$, with polynomial terms independent of $K$ omitted.

[3]Efroni et al. [10] use $\mathcal{N}$, the maximum number of non-zero transition probabilities across the entire state-action space, in their regret and constraint violation analysis. For consistency, we use $|\mathcal{S}|^2|\mathcal{A}|$ to bound $\mathcal{N}$.

[4]Zheng et al. [30] assumes a known transition kernel and analyzes regret in the long-term average setting.

[5]The detailed constraint violation is $\mathcal{O}\left(C''H + H^2\sqrt{|\mathcal{S}|^3|\mathcal{A}|C''\log(C''/\delta')}\right)$, which is independent of $K$. Here, $\delta' = \delta/(16|\mathcal{S}|^2|\mathcal{A}|H)$ and $C'' = \mathcal{O}(\frac{H^4|\mathcal{S}|^3|\mathcal{A}|}{(\tau-c^0)^2}\log\frac{H^4|\mathcal{S}|^3|\mathcal{A}|}{(\tau-c^0)^2\delta'})$.

ensuring a bounded safety constraint violation during learning and a reward regret of $\tilde{\mathcal{O}}(\frac{H^3}{\tau - c^0}\sqrt{|\mathcal{S}|^3|\mathcal{A}|K})$ with respect to the objective.

Compared with the other methods listed in Table 1, though the proposed algorithms have an additional $H/(\tau - c^0)$ or $\sqrt{|\mathcal{S}|}/(\tau - c^0)$ factor in the regret bounds, they are able to reduce the constraint violation to zero or constant with high probability. This improvement in safety can be extremely important for many mission-critical applications.

## 1.1 Related work

The problem of learning an optimal control policy that satisfies safety constraints has been studied both in the RL setting and the multi-armed bandits setting.

**Constrained RL:** Several policy-gradient algorithms have seen success in practice [26, 24, 21, 16, 1, 29]. Also of interest are works which utilize Gaussian processes to model the transition probabilities and value functions [5, 27, 15, 7].

Several algorithms with provable guarantees are closely related to our work. Zheng et al. [30] considered the constrained RL problem in an infinite horizon setting with unknown reward and cost functions. The approach is similar to the UCRL2 algorithm [13] and achieves a sub-linear reward regret $\tilde{\mathcal{O}}(T^{\frac{3}{4}})$ while satisfying constraints with high probability during learning. In contrast to this work, we consider the setting of unknown transition probabilities. Efroni et al [10] focused on the episodic setting of unknown non-stationary transitions over a finite horizon, attaining both a reward regret and a constraint violation of $\tilde{\mathcal{O}}(H^2\sqrt{|\mathcal{S}|^3|\mathcal{A}|K})$. Ding et al. [8] studied an episodic setting with linear function approximation (suitable for large state space cases), and proposed algorithms that can achieve $\tilde{\mathcal{O}}(dH^3\sqrt{K})$ for both the regret and the constraint violation (where $d$ is the dimension of the feature mapping). The regret analysis in [8] can be easily extended to the tabular case, yielding $\tilde{\mathcal{O}}(H^3\sqrt{|\mathcal{S}|^2|\mathcal{A}|K})$.

**Constrained Multi-Armed Bandits:** Multi-armed bandit problems are special cases of MDPs, with both the number of states as well as the episode length being one. Linear bandits with constraints (satisfied with high probability) have been investigated in different settings. One setting, referred to as conservative bandits [28, 14, 11], requires the cumulative reward to remain above a fixed percentage of the cumulative reward of a given baseline policy. Another setting is where each arm is associated with two unknown distributions (similar to our setting), generating reward and cost signals respectively [3, 20, 17, 18].

## 2 Problem formulation

A finite-horizon constrained non-stationary MDP model is defined as a tuple $M = (\mathcal{S}, \mathcal{A}, H, P, r, c, \tau, \mu)$, where $\mathcal{S}$ is the state space, $\mathcal{A}$ is the action space, $H$ is the number of steps in each episode, $r : \mathcal{S} \times \mathcal{A} \to [0, 1]$ is the unknown reward function of interest, $c : \mathcal{S} \times \mathcal{A} \to [0, 1]$ is the unknown safety cost function used to model the constraint violation, $\tau \in (0, H]$ is the given constant used to define the safety constraint, and $\mu$ is the known initial distribution of the state. $P_.(\cdot|s, a) \in \Delta_\mathcal{S}^H, \forall s \in \mathcal{S}, \forall a \in \mathcal{A}$, where $\Delta_\mathcal{S}$ is the $|\mathcal{S}|$-dimensional probability simplex, and $P_h(s'|s, a)$ is the unknown transition probability that the next state is $s'$ when action $a$ is taken for state $s$ at step $h$.

Some further notation is necessary to define the problem. A (randomized Markov) policy is defined by a map $\pi : \mathcal{S} \times [H] \to \Delta_\mathcal{A}$, with $\pi_h(a|s)$ being the probability of taking action $a$ in state $s$ at time step $h$. With $S_t$ and $A_t$ representing the state and the action at time $t$ respectively, let

$$V_h^\pi(s; g, P) := \mathbb{E}_{P, \pi}\left[\sum_{t=h}^{H} g(S_t, A_t)|S_h = s\right], \quad \forall s \in \mathcal{S}$$

denote the expected cumulative value with respect to a function $g : \mathcal{S} \times \mathcal{A} \to \mathbb{R}_+$ under $P$ for a policy $\pi$ over a time interval $[h, h + 1, \ldots, H]$. With slightly abuse of notation, we use $V_1^\pi(\mu; g, P)$ to denote $\mathbb{E}_{S_1 \sim \mu}[V_1^\pi(S_1; g, P)]$.

In the formulation below there are a total of $K$ episodes with $H$ steps each. Each episode $k \in [K]$ begins with an initial probability distribution $\mu$ for $\mathcal{S}_1$. Then, the agent determines a randomized Markov policy $\pi^k$ for that episode based on the information gathered from the previous episodes, and executes it. At time step $h$ during the execution of the $k$-th episode, after taking action $A_h^k$ at state $S_h^k$, the agent receives a noisy reward and cost of $R_h^k(S_h^k, A_h^k) = r_h(S_h^k, A_h^k) + \xi_h^k(S_h^k, A_h^k; r)$ and $C_h^k(S_h^k, A_h^k) = c_h(S_h^k, A_h^k) + \xi_h^k(S_h^k, A_h^k; c)$, respectively.

**Assumption 2.1** (Sub-Gaussian noise). *For all $h \in [H], k \in [K]$, the reward and cost noise random variables are conditionally independent zero-mean 1/2-sub-Gaussian, i.e., $\mathbb{E}[\xi_h^k|\mathcal{F}_{k-1}] = 0$, $\mathbb{E}[\exp(\lambda\xi_h^k)|\mathcal{F}_{k-1}] \leq \exp(\lambda^2/4), \forall \lambda \in \mathbb{R}$. Here $\mathcal{F}_k$ is the $\sigma$-algebra generated by the random variables up to episode $k$.*

Let $\pi^*$ denote the optimal policy of the following CMDP model:

$$\max_\pi \quad V_1^\pi(\mu; r, P) \qquad \text{s.t.} \quad V_1^\pi(\mu; c, P) \leq \tau. \tag{1}$$

A policy $\pi$ is said to be *strictly safe* if $V_1^\pi(\mu; c, P) < \tau$.

**Assumption 2.2.** *There exists a strictly safe policy $\pi^0$ with $V_1^{\pi^0}(\mu; c, P) = c^0 < \tau$.*

There are two important cases of the safe learning problem.

**Zero constraint violation case**: The agent has prior knowledge of a strictly safe policy $\pi^0$ and its safety cost value $c^0 := V_1^{\pi^0}(\mu; c, P)$. The agent wishes to attain a sublinear (in $K$) *cumulative regret*,

$$Reg(K; r) := \sum_{k=1}^K \left( V_1^{\pi^*}(\mu; r, P) - V_1^{\pi^k}(\mu; r, P) \right), \tag{2}$$

while incurring zero constraint violation with at least a specified high probability $(1 - \delta)$, i.e.,

$$\mathbb{P}\left( V_1^{\pi^k}(\mu; c, P) \leq \tau, \forall k \in [K] \right) \geq 1 - \delta.$$

**Bounded constraint violation case**: The agent knows that there exists a strictly safe policy with a known safety cost value $c^0$, but does not know any strictly safe policy. It aims to achieve a cumulative regret (2) that grows sublinearly with $K$, while ensuring that the *regret of constraint violation*,

$$Reg(K; c) := \left( \sum_{k=1}^K \left( V_1^{\pi^k}(\mu; c, P) - \tau \right) \right)_+ \quad (\text{where } (a)_+ := \max\{a, 0\}),$$

satisfies $\sup_K Reg(K; c) < +\infty$ with at least a specified high probability $(1 - \delta)$.

**Remarks.** For the zero constraint violation case, the assumption of knowing $c^0$ can be relaxed, as shown in Appendix E.

## 3 Zero constraint violation case

We start by considering the zero constraint violation case. On one hand, to balance the exploration-exploitation trade-off, we employ an optimistic estimate of the reward function, as embodied in the OFU principle. On the other hand, to maintain absolute safety with high probability during the exploration, we employ a pessimistic estimate of the safety cost. Such an OPFU principle was previously discussed in constrained bandits [3] and we adapt it to the unknown CMDP setting.

At each episode $k$, we begin by forming empirical estimates of the transition probabilities, the reward function, and the cost function from step $h$ of all previous episodes:

$$\hat{P}_h^k(s'|s, a) := \frac{\sum_{k'=1}^{k-1} \mathbb{1}(S_h^{k'} = s, A_h^{k'} = a, S_{h+1}^{k'} = s')}{N_h^k(s, a) \vee 1}, \tag{3}$$

$$\hat{g}_h^k(s, a) := \frac{\sum_{k'=1}^{k-1} \mathbb{1}(S_h^{k'} = s, A_h^{k'} = a)(g_h(s, a) + \xi_h^k(s, a; g))}{N_h^k(s, a) \vee 1}, \quad \text{for } g = r, c, \tag{4}$$

where $a \vee b := \max\{a, b\}$, and

$$N_h^k(s, a) := \# \text{ of visits to state-action pair } (s, a) \text{ at step } h \text{ in episodes } [1, 2, \ldots, k - 1]. \quad (5)$$

Next we fix some $\delta \in (0, 1)$ and form a common (for notational simplicity) confidence radius $\beta_h^k(s, a)$ for the transition probabilities, the rewards, and the costs,

$$\beta_h^k(s, a) := \sqrt{\frac{1}{N_h^k(s, a) \vee 1} Z}, \quad \text{where } Z := \log(16|\mathcal{S}|^2 |\mathcal{A}| HK/\delta).$$

At each episode $k$, we define the *optimistically biased reward estimate* as

$$\bar{r}_h^k(s, a) := \hat{r}_h^k(s, a) + \alpha_r \beta_h^k(s, a), \quad \forall (s, a, h) \in \mathcal{S} \times \mathcal{A} \times [H], \quad (6)$$

where the scaling factor is

$$\alpha_r := 1 + |\mathcal{S}|H + \frac{4H(1 + |\mathcal{S}|H)}{\tau - c^0}. \quad (7)$$

To guarantee safe exploration, define the *pessimistically biased safety cost estimate* at episode $k$ as

$$\underline{c}_h^k(s, a) := \hat{c}_h^k(s, a) + (1 + H|\mathcal{S}|)\beta_h^k(s, a), \quad \forall (s, a, h) \in \mathcal{S} \times \mathcal{A} \times [H]. \quad (8)$$

The policy we execute at episode $k$ is chosen from a "pessimistically safe" policy set $\Pi^k$, defined as

$$\Pi^k := \begin{cases} \{\pi^0\} & \text{if } V_1^{\pi^0}(\mu; \underline{c}^k, \hat{P}^k) \geq (\tau + c^0)/2, \\ \{\pi : V_1^\pi(\mu; \underline{c}^k, \hat{P}^k) \leq \tau\} & \text{otherwise.} \end{cases} \quad (9)$$

We simply use the strictly safe policy $\pi^0$ until $V_1^{\pi^0}(\mu; \underline{c}^k, \hat{P}^k) < (\tau + c^0)/2$, which is a sufficient condition to guarantee that the set $\{\pi : V_1^\pi(\mu; \underline{c}^k, \hat{P}^k) \leq \tau\}$ is non-empty. Within this set, we choose the optimistically best reward earning policy $\pi^k$, which can be solved by linear programming with $\Theta(|\mathcal{S}||\mathcal{A}|H)$ decision variables and constraints [10]. The resulting Optimistic Pessimism-based Linear Programming (OptPess-LP) algorithm is presented below:

---
**Algorithm 1: OptPess-LP**

---
**Input:** $K, \delta, \pi^0, c^0, \tau$;
**Initialization:** $N_h^1(s, a) = 0, \ \forall (s, a, h) \in \mathcal{S} \times \mathcal{A} \times [H]$;
**for** $k = 1, 2, \ldots, K$ **do**

    Update empirical model (i.e., $\hat{P}^k, \hat{r}^k, \hat{c}^k$) as in Equations (3)-(5);
    Update $\bar{r}^k, \underline{c}^k$, and $\Pi^k$ as in Equations (6)-(9);
    Calculate $\pi^k \in \arg\max_{\pi \in \Pi^k} V_1^\pi(\mu; \bar{r}^k, \hat{P}^k)$;
    Execute $\pi^k$ and collect a trajectory $(S_h^k, A_h^k, R_h^k, C_h^k), \ \forall h \in [H]$;
    Update counters $N_h^{k+1}(S_h^k, A_h^k), \ \forall h \in [H]$;
**end**

---

**Theorem 3.1** (Regret and constraint violation bounds for OptPess-LP). *Fix any $\delta \in (0, 1)$. With probability at least $(1 - \delta)$, OptPess-LP has zero constraint violation with*

$$Reg^{OPLP}(K; r) = \tilde{\mathcal{O}}\left(\frac{H^3}{\tau - c^0}\sqrt{|\mathcal{S}|^3 |\mathcal{A}| K} + \frac{H^5 |\mathcal{S}|^3 |\mathcal{A}|}{(\tau - c^0)^2 \wedge (\tau - c^0)}\right), \quad \text{where } a \wedge b := \min\{a, b\}.$$

Theorem 3.1 shows that it is possible to achieve sublinear regret in $K$, while simultaneously incurring no constraint violation with arbitrarily high probability. The proof is sketched in Section 5.1, with detailed proofs presented in Appendix B.

## 4 Bounded constraint violation case

Without prior knowledge of a strictly safe policy $\pi^0$, we may not be able to guarantee zero constraint violation with high probability. However, by relaxing the requirement to bounded (in $K$) safety constraint violation, we can incorporate more exploration and design a more efficient algorithm by a primal-dual approach. It is inspired by the design of the pessimistic term in constrained bandits [17].

Different from traditional optimistic dual analysis methods in [10], we introduce an additive pessimistic term $\epsilon_k$ at each episode $k$ in the original optimization problem (1), i.e.,

$$\max_\pi \quad V_1^\pi(\mu; r, P) \qquad \text{s.t.} \quad V_1^\pi(\mu; c, P) + \epsilon_k \leq \tau. \tag{10}$$

The pessimistic term restrains the constraint violation and will be progressively decreased as learning proceeds.

The CMDP problem in (10) may however not have any feasible solution. To overcome this, we consider the Lagrangian,

$$L^k(\pi, \lambda) := V_1^\pi(\mu; r, P) + \lambda\left(\tau - \epsilon_k - V_1^\pi(\mu; c, P)\right),$$

for which, given any Lagrange multiplier $\lambda$, we can always solve for the optimizer $\max_\pi L^k(\pi, \lambda)$ by dynamic programming.

With the introduction of the additive term $\epsilon_k$, the dual variable (denoted by $\lambda^k$) governed by the subgradient algorithm grows faster, which enhances safety constraints in the next episode.

In order to guarantee sufficient optimism of rewards and costs, we integrate the uncertainty of transitions into rewards and costs, i.e.,

$$\tilde{r}_h^k(s, a) := \hat{r}_h^k(s, a) + \beta(n_k(s, a, h)) + H|\mathcal{S}|\beta(n_k(s, a, h)), \quad \forall (s, a, h) \in \mathcal{S} \times \mathcal{A} \times [H],$$
$$\tilde{c}_h^k(s, a) := \hat{c}_h^k(s, a) - \beta(n_k(s, a, h)) - H|\mathcal{S}|\beta(n_k(s, a, h)), \quad \forall (s, a, h) \in \mathcal{S} \times \mathcal{A} \times [H]. \tag{11}$$

Note that in contrast to the zero constraint violation case, we *optimistically* estimate the safety cost function, with the pessimism only governed by $\epsilon_k$.

We employ truncated value functions due to the additional uncertainties from transitions, i.e.,

$$\hat{V}_1^\pi(\mu; \tilde{r}^k, \hat{P}^k) := \min\{H, V_1^\pi(\mu; \tilde{r}^k, \hat{P}^k)\}, \quad \hat{V}_1^\pi(\mu; \tilde{c}^k, \hat{P}^k) := \max\{0, V_1^\pi(\mu; \tilde{c}^k, \hat{P}^k)\}. \tag{12}$$

For the policy update of the primal variable (denoted by $\pi^k$), we can apply standard dynamic programming by viewing $\tilde{r}_h^k(s, a) - \frac{\lambda^k}{\eta^k}(\tilde{c}_h^k(s, a) - \tau)$ as the reward function. Specifically, we apply backward induction to solve for the optimal policy:

$$Q_h^k(s, a) = \tilde{r}_h^k(s, a) - \frac{\lambda^k}{\eta^k}(\tilde{c}_h^k(s, a) - \tau) + \sum_{s' \in \mathcal{S}} \hat{P}_h^k(s'|s, a) \max_{a'} Q_{h+1}^k(s', a'), \quad \forall h \in [H]$$

with $Q_{H+1}^k(s, a) = 0, \forall (s, a) \in \mathcal{S} \times \mathcal{A}$. Then, $\pi_h^k \in \arg\max_a Q_h^k(s, a)$, which is computationally efficient (as efficient as policy-gradient-based algorithms in the tabular case).

The resulting Optimistic Pessimism-based Primal-Dual (OptPess-PrimalDual) algorithm is shown in Algorithm 2. It chooses $\epsilon_k := 5H^2\sqrt{|\mathcal{S}|^3|\mathcal{A}|}(\log\frac{k}{\delta'} + 1)/\sqrt{k\log\frac{k}{\delta'}}$, $\delta' = \delta/(16|\mathcal{S}|^2|\mathcal{A}|H)$, the scaling parameter $\eta^k := (\tau - c^0)H\sqrt{k}$ in the primal policy update, and, for convenience, a step size of 1 for the dual update.

**Theorem 4.1** (Regret and constraint violation bounds for OptPess-PrimalDual). *Fix any $\delta \in (0, 1)$. Then,*

$$Reg^{OPPD}(K; r) = \tilde{\mathcal{O}}\left(\frac{H^3}{\tau - c^0}\sqrt{|\mathcal{S}|^3|\mathcal{A}|K} + \frac{H^5|\mathcal{S}|^3|\mathcal{A}|}{(\tau - c^0)^2}\right),$$

$$Reg^{OPPD}(K; c) = \mathcal{O}\left(C''(H - \tau) + H^2\sqrt{|\mathcal{S}|^3|\mathcal{A}|C''}\right) = \mathcal{O}(1),$$

*where $C'' = \mathcal{O}(\frac{H^4|\mathcal{S}|^3|\mathcal{A}|}{(\tau - c^0)^2}\log\frac{H^4|\mathcal{S}|^3|\mathcal{A}|}{(\tau - c^0)^2\delta'})$ is a coefficient independent of $K$.*

Theorem 4.1 shows that it is possible to achieve an $\tilde{\mathcal{O}}(\sqrt{K})$ reward regret, while only allowing bounded constraint violation with arbitrarily high probability. Detailed proofs are presented in Section 5.2 and Appendix C.

**Algorithm 2: OptPess-PrimalDual**

---

**Input:** $K, \delta, c^0, \tau$;
**Initialization:** $N_h^1(s,a) = 0$, $\forall (s,a,h) \in \mathcal{S} \times \mathcal{A} \times [H]$, $\lambda^1 = 0$;
**for** $k = 1, 2, \ldots, K$ **do**

> Set $\epsilon_k = 5H^2\sqrt{|\mathcal{S}|^3|\mathcal{A}|}(\log\frac{k}{\delta'}+1)/\sqrt{k\log\frac{k}{\delta'}}$, $\delta' = \delta/(16|\mathcal{S}|^2|\mathcal{A}|H)$,
>
> $\quad \eta^k = (\tau - c^0)H\sqrt{k}$;
>
> Update empirical model (i.e., $\hat{P}^k, \hat{r}^k, \hat{c}^k$) as in Equations (3)-(5);
> Update $\tilde{r}^k, \tilde{c}^k$ as in Equation (11);
>
> *(Policy Update)* $\pi^k \in \arg\max_{\pi \in \Pi} \hat{V}_1^\pi(\mu; \tilde{r}^k, \hat{P}^k) - \frac{\lambda^k}{\eta^k}\left(\hat{V}_1^\pi(\mu; \tilde{c}^k, \hat{P}^k) - \tau\right)$;
>
> *(Dual Update)* $\lambda^{k+1} = \left(\lambda^k + \hat{V}_1^{\pi^k}(\mu; \tilde{c}^k, \hat{P}^k) + \epsilon_k - \tau\right)_+$;
>
> Execute $\pi^k$ and collect a trajectory $(S_h^k, A_h^k, R_h^k, C_h^k)$, $\forall h \in [H]$;
> Update counters $N_h^{k+1}(S_h^k, A_h^k)$, $\forall h \in [H]$;

**end**

---

# 5 Regret and constraint violation analysis

We sketch the key steps in the proofs of Theorems 3.1 and 4.1. Full details are relegated to Appendices B and C, respectively. Note that our analysis and results are conditioned on the same high probability event (specifically defined in Appendix A), which occurs with probability at least $(1 - \delta)$.

## 5.1 Analysis of OptPess-LP (Algorithm 1)

**Constraint violation analysis** The zero constraint violation of OptPess-LP follows from the following property of the pessimistic policy set $\Pi^k$:

**Lemma 5.1.** *With probability at least $(1-\delta)$, for any $k \in [K]$ and policy $\pi \in \Pi^k$, $V_1^\pi(\mu; c, P) \leq \tau$.*

**Regret of reward analysis** When the parameters $c, P$ are not well estimated, there may not exist any policy $\pi$ such that $V_1^\pi(\mu; \underline{c}^k, \hat{P}^k) \leq \tau$. Hence, as defined in (9), $\Pi^k$ is a singleton set $\{\pi^0\}$, and accordingly $\pi^0$ is executed, even though $\pi^0$ is not safe for $(\underline{c}^k, \hat{P}^k)$. It subsequently takes several episodes of exploration using $\pi^0$ until it becomes strictly safe for $(\underline{c}^k, \hat{P}^k)$. At that time, policies close enough to $\pi^0$, of which there are infinitely many, are also safe for $(\underline{c}^k, \hat{P}^k)$, and so $|\Pi^k| = +\infty$. At this point the learning algorithm can proceed to enhance reward while maintaining safety with respect to $(\underline{c}^k, \hat{P}^k)$.

To analyze the algorithm, we decompose the reward regret as follows:

$$
\begin{aligned}
Reg^{\textbf{OPLP}}(K; r) = &\sum_{k=1}^K \mathbb{1}(|\Pi^k| = 1)\left(V_1^{\pi^*}(\mu; r, P) - V_1^{\pi^0}(\mu; r, P)\right) \\
&+ \sum_{k=1}^K \mathbb{1}(|\Pi^k| > 1)\left(V_1^{\pi^*}(\mu; r, P) - V_1^{\pi^k}(\mu; \bar{r}^k, \hat{P}^k)\right) \\
&+ \sum_{k=1}^K \mathbb{1}(|\Pi^k| > 1)\left(V_1^{\pi^k}(\mu; \bar{r}^k, \hat{P}^k) - V_1^{\pi^k}(\mu; r, P)\right). \quad (13)
\end{aligned}
$$

To bound the first term on the right-hand side (RHS) of (13), we have the following lemma which gives an upper bound on the number of episodes for exploration by policy $\pi^0$:

**Lemma 5.2.** *With probability at least $(1 - \delta)$, $\sum_{k=1}^K \mathbb{1}(|\Pi^k| = 1) \leq C'$, where $C' = \tilde{\mathcal{O}}(H^4|\mathcal{S}|^3|\mathcal{A}|/((\tau - c^0)^2 \wedge (\tau - c^0)))$.*

Turning to the second term on the RHS of (13), we first note that $\pi^*$ may not be in $\Pi^k$. To ensure that the term $V_1^{\pi^*}(\mu; r, P) - V_1^{\pi^k}(\mu; \bar{r}^k, \hat{P}^k)$ is nevertheless non-positive even when $\pi^* \notin \Pi^k$, we set $\alpha_r$

to the large value shown in (7). This increases $\bar{r}^k$, and hence also $V_1^{\pi^k}(\mu; \bar{r}^k, \hat{P}^k)$. In addition, we show that there is a policy $\hat{\pi}^k$ that attains the same reward as the (non-Markov) probabilistic mixed policy, $\tilde{\pi}^k := B_{\gamma_k}\pi^* + (1 - B_{\gamma_k})\pi^0$, where $B_{\gamma_k}$ is a Bernoulli distributed random variable with mean $\gamma_k$ for $\gamma_k \in [0, 1]$. $\gamma_k$ will be chosen as the largest coefficient such that $V_1^{\tilde{\pi}^k}(\mu; \underline{c}^k, \hat{P}^k) \leq \tau$. This latter policy in turn has a larger reward than $\pi^0$ since it is a mixture with $\pi^*$, yielding the following:

**Lemma 5.3.** *With probability at least* $(1 - \delta)$,

$$\sum_{k=1}^{K} \mathbb{1}(|\Pi^k| > 1)\left(V_1^{\pi^*}(\mu; r, P) - V_1^{\pi^k}(\mu; \bar{r}^k, \hat{P}^k)\right) \leq 0.$$

Finally, concerning the third term on the RHS of (13), akin to the closed-loop identifiability property [6, 19], while $\hat{P}^k$ may not converge to $P$, the difference in the rewards $V_1^{\pi^k}(\mu; \bar{r}^k, \hat{P}^k) - V_1^{\pi^k}(\mu; r, P)$ grows sublinearly in $k$ since the same policy $\pi^k$ is used in both values:

**Lemma 5.4.** *With probability at least* $(1 - \delta)$,

$$\sum_{k=1}^{K} \mathbb{1}(|\Pi^k| > 1)\left(V_1^{\pi^k}(\mu; \bar{r}^k, \hat{P}^k) - V_1^{\pi^k}(\mu; r, P)\right) = \tilde{\mathcal{O}}\left(\frac{H^3}{\tau - c^0}\sqrt{|\mathcal{S}|^3|\mathcal{A}|K} + \frac{H^5|\mathcal{S}|^3|\mathcal{A}|}{\tau - c^0}\right).$$

Combining Lemmas 5.2, 5.3, and 5.4 yields Theorem 3.1.

## 5.2 Analysis of OptPess-PrimalDual (Algorithm 2)

In this section, we outline the steps in the proof of Theorem 4.1 by analyzing regret and constraint violation of OptPess-PrimalDual respectively.

Recall $\epsilon_k = 5H^2\sqrt{|\mathcal{S}|^3|\mathcal{A}|}(\log \frac{k}{\delta'} + 1)/\sqrt{k \log \frac{k}{\delta'}}$, where $\delta' = \delta/(16|\mathcal{S}|^2|\mathcal{A}|H)$. The existence of a feasible solution to (10) can be guaranteed if $\epsilon_k \leq \tau - c^0$. Let $C''$ be the smallest value such that $\forall k \geq C''$, $\epsilon_k \leq (\tau - c^0)/2$. Then the perturbed optimization problem (10) has at least one feasible solution for any $k \geq C''$. By simple calculation, one can verify that $C'' = \mathcal{O}(\frac{H^4|\mathcal{S}|^3|\mathcal{A}|}{(\tau - c^0)^2}\log\frac{H^4|\mathcal{S}|^3|\mathcal{A}|}{(\tau - c^0)^2\delta'})$. Notice that $\epsilon_k$ is a function not depending on $K$, and so is the coefficient $C''$.

**Constraint violation analysis** The bounded constraint violation of OptPess-PrimalDual is established as follows. We first decompose the constraint violation as

$$Reg^{\mathbf{OPPD}}(K; c) = \left(\sum_{k=1}^{K}\left(V_1^{\pi^k}(\mu; c, P) - \hat{V}_1^{\pi^k}(\mu; \tilde{c}^k, \hat{P}^k)\right) + \sum_{k=1}^{K}\left(\hat{V}_1^{\pi^k}(\mu; \tilde{c}^k, \hat{P}^k) - \tau\right)\right)_+$$

$$\leq \left(\sum_{k=1}^{K}\left(V_1^{\pi^k}(\mu; c, P) - \hat{V}_1^{\pi^k}(\mu; \tilde{c}^k, \hat{P}^k)\right) + \lambda^{K+1} - \sum_{k=1}^{K}\epsilon_k\right)_+. \quad (14)$$

The first summation term and $\lambda^{K+1}$ in (14) can be bounded as follows:

**Lemma 5.5.** *Recall* $\delta' = \delta/(16|\mathcal{S}|^2|\mathcal{A}|H)$, *with probability at least* $(1 - \delta)$,

$$\sum_{k=1}^{K}\left(V_1^{\pi^k}(\mu; c, P) - \hat{V}_1^{\pi^k}(\mu; \tilde{c}^k, \hat{P}^k)\right) \leq 8H^2\sqrt{|\mathcal{S}|^3|\mathcal{A}|K\log\frac{K}{\delta'}} + \mathcal{O}(PolyLog(K)).$$

**Lemma 5.6.** *For any* $k \geq C''$, *with probability at least* $(1 - \delta)$,

$$\lambda^k \leq \frac{1}{\zeta}\ln\frac{11\nu_{\max}^2}{3\rho^2} + C''(H - \tau) + \sum_{u=1}^{C''}\epsilon_u + H + \frac{4(H^2 + \epsilon_k^2 + \eta^k H)}{\tau - c^0},$$

*where* $\rho = (\tau - c^0)/4$, $\nu_{\max} = H$, $\zeta = \rho/(\nu_{\max}^2 + \nu_{\max}\rho/3)$.

To guarantee bounded violation, we ensure that $\sum_{k=1}^{K} \epsilon_k$ in (14) can cancel the dominant terms in the two lemmas above. According to Lemmas 5.5 and 5.6, with probability at least $(1 - \delta)$, the violation is bounded as

$$Reg^{\textbf{OPPD}}(K; c) = \mathcal{O}\left( C'' H + H^2 \sqrt{|\mathcal{S}|^3 |\mathcal{A}| C'' \log\left( C''/\delta' \right)} \right).$$

**Regret of reward analysis**   For episode $k$ with $k \geq C''$, let $\pi^{\epsilon_k, *}$ be the optimal policy for (10), which is well-defined by the definition of $C''$. We decompose the reward regret as

$$Reg^{\textbf{OPPD}}(K; r) = \sum_{k=1}^{C''} \left( V_1^{\pi^*}(\mu; r, P) - V_1^{\pi^k}(\mu; r, P) \right) \tag{15}$$

$$+ \sum_{k=C''}^{K} \left( V_1^{\pi^*}(\mu; r, P) - V_1^{\pi^{\epsilon_k, *}}(\mu; r, P) \right) + \sum_{k=C''}^{K} \left( V_1^{\pi^{\epsilon_k, *}}(\mu; r, P) - \hat{V}_1^{\pi^{\epsilon_k, *}}(\mu; \tilde{r}^k, \hat{P}^k) \right)$$

$$+ \sum_{k=C''}^{K} \left( \hat{V}_1^{\pi^{\epsilon_k, *}}(\mu; \tilde{r}^k, \hat{P}^k) - \hat{V}_1^{\pi^k}(\mu; \tilde{r}^k, \hat{P}^k) \right) + \sum_{k=C''}^{K} \left( \hat{V}_1^{\pi^k}(\mu; \tilde{r}^k, \hat{P}^k) - V_1^{\pi^k}(\mu; r, P) \right).$$

We upper bound each term on the RHS of (15). Since $V_1^{\pi}(\mu; r, P) \in [0, H]$ for any policy $\pi$, the first term is upper bounded by $HC''$. The second and third terms can be bounded by the following two lemmas:

**Lemma 5.7.** *With probability at least $(1 - \delta)$,*

$$\sum_{k=C''}^{K} \left( V_1^{\pi^*}(\mu; r, P) - V_1^{\pi^{\epsilon_k, *}}(\mu; r, P) \right) \leq \sum_{k=C''}^{K} \frac{\epsilon_k H}{\tau - c^0} = \tilde{\mathcal{O}}\left( \frac{H^3}{\tau - c^0} \sqrt{|\mathcal{S}|^3 |\mathcal{A}| K} \right).$$

**Lemma 5.8.** *With probability at least $(1 - \delta)$, $\sum_{k=C''}^{K} \left( V_1^{\pi^{\epsilon_k, *}}(\mu; r, P) - \hat{V}_1^{\pi^{\epsilon_k, *}}(\mu; \tilde{r}^k, \hat{P}^k) \right) \leq 0$.*

The pivotal step is to leverage optimism of $\pi^k$ to further decompose the fourth term on the RHS of (15), and utilize the projected dual update to transfer it into the form of $\lambda^k (\lambda^k - \lambda^{k+1})$. The following lemmas provide high probability bounds for the remaining two terms on the RHS of (15):

**Lemma 5.9.** *With probability at least $(1 - \delta)$,*

$$\sum_{k=C''}^{K} \left( \hat{V}_1^{\pi^{\epsilon_k, *}}(\mu; \tilde{r}^k, \hat{P}^k) - \hat{V}_1^{\pi^k}(\mu; \tilde{r}^k, \hat{P}^k) \right) = \tilde{\mathcal{O}}\left( \frac{H}{\tau - c^0} \sqrt{K} \right).$$

**Lemma 5.10.** *With probability at least $(1 - \delta)$,*

$$\sum_{k=C''}^{K} \left( \hat{V}_1^{\pi^k}(\mu; \tilde{r}^k, \hat{P}^k) - V_1^{\pi^k}(\mu; r, P) \right) = \tilde{\mathcal{O}}\left( H^2 \sqrt{|\mathcal{S}|^3 |\mathcal{A}| K} + H^4 |\mathcal{S}|^3 |\mathcal{A}| \right).$$

Applying Lemmas 5.7, 5.8, 5.9, and 5.10 yields Theorem 4.1.

## 6   Concluding remarks

We present two optimistic pessimism-based algorithms that maintain stringent safety constraints (either zero or bounded safety constraint violation) for unknown CMDPs with high probability, while still attaining an $\tilde{\mathcal{O}}(\sqrt{K})$ regret of reward. The algorithms employ, respectively, a pessimistically safe policy set $\Pi^k$ or an additional pessimistic term $\epsilon_k$ into the safety constraint. The first algorithm, OptPess-LP, guarantees zero violation with high probability by solving a linear programming with $\Theta(|\mathcal{S}||\mathcal{A}|H)$ decision variables, while the second algorithm, OptPess-PrimalDual, is as efficient as policy-gradient-based algorithms in the tabular case, but violates constraints during initial episodes. A possible future direction for exploration is the application of the above OPFU principle in model-free algorithms.

## Acknowledgement

P. R. Kumar's work is partially supported by US National Science Foundation under CMMI-2038625, HDR Tripods CCF-1934904; US Office of Naval Research under N00014-21-1-2385; US ARO under W911NF1810331, W911NF2120064; and U.S. Department of Energy's Office of Energy Efficiency and Renewable Energy (EERE) under the Solar Energy Technologies Office Award Number DE-EE0009031. The views expressed herein and conclusions contained in this document are those of the authors and should not be interpreted as representing the views or official policies, either expressed or implied, of the U.S. NSF, ONR, ARO, Department of Energy or the United States Government. The U.S. Government is authorized to reproduce and distribute reprints for Government purposes notwithstanding any copyright notation herein.

Dileep Kalathil gratefully acknowledges funding from the U.S. National Science Foundation (NSF) grants NSF-CRII- CPS-1850206 and NSF-CAREER-EPCN-2045783.

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
