# Appendix

The theorems and lemmas presented in the paper are provided with full details in this appendix.

First, let's recall some notations. Fix some $0 < \delta < 1$ as the input of algorithms. We say $g = \tilde{\mathcal{O}}(f)$ if there exists a universal constant $C$ such that $g \leq C \left( f \log \frac{1}{\delta} + f \log f \right)$ for any $f$ and $\delta$. The filtration $\{\mathcal{F}_k\}_{k \geq 0}$ is defined as follows: $\mathcal{F}_0$ is the trivial sigma algebra, and for each $k \in [K]$, $\mathcal{F}_k = \sigma \left( (S_h^{k'}, A_h^{k'}, R_h^{k'}, C_h^{k'})_{h \in [H], k' \in [k]} \right)$. The policy process, i.e., the sequence of deployed policies $\{\pi^k\}_{k \in [K]}$, is a predictable process with respect to the filtration $\{\mathcal{F}_k\}_{k \geq 0}$. According to the definition of $N_h^k$, we know $N_h^k(s, a) \in \mathcal{F}_{k-1}$.

Additionally, we define expectation operator $\mathbb{E}_{\mu', P', \pi'}[\cdot]$ as the expectation with respect to a stochastic trajectory $(S_h, A_h)_{h \in [H]}$ generated according to the Markov chain induced by $(\mu', P', \pi')$. When $\mu', P', \pi'$ are random elements, $\mathbb{E}_{\mu', P', \pi'}[\cdot]$ will be a $\sigma(\mu', P', \pi')$-measurable random variable.

## A    High probability good event $\mathcal{E}$

We aim to give performance guarantees of the algorithms with high probability. To this end, we first consider a high probability "good event" $\mathcal{E}$ that all the subsequent analysis is conditioned on.

First, for any predictable event sequence $\mathcal{G}_{1:K}$, i.e., $\mathcal{G}_k \in \mathcal{F}_{k-1}, \forall k \in [K]$, define the event

$$
\mathcal{E}_{\mathcal{G}}(\delta) := \Bigg\{ \forall K' \in [K], \; \sum_{k=1}^{K'} \sum_{h=1}^{H} \sum_{s,a} \frac{\mathbb{1}(\mathcal{G}_k) q^{\pi^k}(s, a, h)}{N_h^k(s, a) \vee 1} \leq 4H|\mathcal{S}||\mathcal{A}| + 2H|\mathcal{S}||\mathcal{A}| \ln K'_{\mathcal{G}} + 4 \ln \frac{2HK}{\delta},
$$

$$
\sum_{k=1}^{K'} \sum_{h=1}^{H} \sum_{s,a} \frac{\mathbb{1}(\mathcal{G}_k) q^{\pi^k}(s, a, h)}{\sqrt{N_h^k(s, a) \vee 1}} \leq 6H|\mathcal{S}||\mathcal{A}| + 2H\sqrt{|\mathcal{S}||\mathcal{A}|K'_{\mathcal{G}}} + 2H|\mathcal{S}||\mathcal{A}| \ln K'_{\mathcal{G}} + 5 \ln \frac{2HK}{\delta} \Bigg\},
$$

where $K'_{\mathcal{G}} := \sum_{k=1}^{K'} \mathbb{1}(\mathcal{G}_k)$ and $q^{\pi^k}$ is the occupancy measure of policy $\pi^k$, i.e., $q^{\pi^k}(s, a, h) = \mathbb{E}_{\mu, P, \pi^k}[\mathbb{1}(S_h^k = s, A_h^k = a)|\mathcal{F}_{k-1}]$.

A trivial predictable event sequence is $\mathcal{G}_{1:k}$ with $\mathcal{G}_k = \Omega, \forall k \in [K]$, where $\Omega$ is the sample space. Let $\mathcal{E}_{\Omega}(\delta)$ be the event defined by this trivial event sequence.

Consider another event sequence $\mathcal{G}'_{1:K} = \{V_1^{\pi^0}(\mu; \underline{c}^k, \hat{P}^k) \geq \frac{\tau + c^0}{2}\}_{k \in [K]}$, which is predictable with respect to $\{\mathcal{F}_k\}_{k \geq 0}$. Let $\mathcal{E}_0(\delta)$ be the event $\mathcal{E}_{\mathcal{G}'}(\delta)$ defined by this event sequence $\mathcal{G}'_{1:K}$. Notice that all notations (including $\hat{r}, \hat{c}, \hat{P}$, and $\underline{c}^k$) are defined in the same manner in the two algorithms, so this event sequence $\mathcal{G}'_{1:K}$ is also well-defined in Algorithm 2.

**Good event $\mathcal{E}$.**    Define a "good event" $\mathcal{E}$ as

$$
\begin{aligned}
\mathcal{E} := \Big\{ &\forall k \in [K], \forall h \in [H], \forall s \in \mathcal{S}, \forall a \in \mathcal{A}, \\
&|r_h(s, a) - \hat{r}_h^k(s, a)| \leq \beta_h^k(s, a), \; |c_h(s, a) - \hat{c}_h^k(s, a)| \leq \beta_h^k(s, a), \\
&|\hat{P}_h^k(s'|s, a) - P_h(s'|s, a)| \leq \beta_h^k(s, a), \forall s' \in \mathcal{S} \\
&|\hat{P}_h^k(s'|s, a) - P_h(s'|s, a)| \leq \tilde{\beta}_h^k(s'|s, a), \forall s' \in \mathcal{S} \Big\} \cap \mathcal{E}_{\Omega}(\delta/4) \cap \mathcal{E}_0(\delta/4), \quad (16)
\end{aligned}
$$

where $\tilde{\beta}_h^k(s'|s, a) := \sqrt{\frac{2P(s'|s,a)}{N_h^k(s,a) \vee 1} Z} + \frac{Z}{3N_h^k(s,a) \vee 1}$ and $Z := \log(16|\mathcal{S}|^2|\mathcal{A}|HK/\delta)$.

**Lemma A.1.** *Fix any $\delta \in (0, 1)$ as the confidence parameter in the inputs of the proposed algorithms. The good event $\mathcal{E}$ occurs with probability at least $1 - \delta$.*

*Proof of Lemma A.1.* For each $(s, a, h) \in \mathcal{S} \times \mathcal{A} \times [H]$, we take $K$ mutually independent samples of the reward, cost, and next state with the probability distribution specified by the generative model $M$:

$$
\{R^n(s, a, h), C^n(s, a, h), S^n(s, a, h)\}_{n=1}^{K},
$$

Let $(\hat{r}^n(s,a,h), \hat{c}^n(s,a,h), \hat{P}^n(\cdot|s,a,h))$ be the corresponding (running) empirical means, respectively, for the samples

$$\{R^i(s,a,h), C^i(s,a,h), S^i(s,a,h)\}_{i=1}^n.$$

Define the following failure events:

$$\begin{aligned}
F_n^r &:= \{\exists s,a,h : |\hat{r}^n(s,a,h) - r_h(s,a)| \geq \beta(n)\}, \\
F_n^c &:= \{\exists s,a,h : |\hat{c}^n(s,a,h) - c_h(s,a)| \geq \beta(n)\}, \\
F_n^P &:= \{\exists s,a,s',h : |P_h(s'|s,a) - \hat{P}^n(s'|s,a,h)| \geq \beta(n)\}, \\
\tilde{F}_n^P &:= \{\exists s,a,s',h : |P_h(s'|s,a) - \hat{P}^n(s'|s,a,h)| \geq \tilde{\beta}(n, P_h(s'|s,a))\},
\end{aligned}$$

where $\beta(n) := \sqrt{\frac{1}{n \vee 1} Z}$ and $\tilde{\beta}(n, p) := \sqrt{\frac{2P(s'|s,a)}{n \vee 1} Z} + \frac{Z}{3n \vee 1}$.

Define the event $\mathcal{E}^{gen}$

$$\mathcal{E}^{gen} := \left( (\bigcup_{n=1}^K F_n^r \cup F_n^c \cup F_n^P \cup \tilde{F}_n^P)^C \cap \mathcal{E}_\Omega(\delta/4) \cap \mathcal{E}_0(\delta/4) \right).$$

By the definition of $\mathcal{E}_\Omega(\delta/4)$ and $\mathcal{E}_0(\delta/4)$ and Lemma D.5, event $\mathcal{E}_\Omega(\delta/4) \cap \mathcal{E}(\delta/4)$ occurs with probability at least $1 - \delta/2$. Therefore, to show that $\mathbb{P}(\mathcal{E}^{gen}) \geq 1 - \delta$, it is sufficient to show that $\mathbb{P}(\bigcup_{n=1}^K F_n^r \cup F_n^c \cup F_n^P \cup \tilde{F}_n^P) \leq \delta/2$. Note that $\delta/16 \leq \delta|\mathcal{S}|/4(1 + |\mathcal{S}|) =: \delta'$. Now, it is straightforward to show the following:

(i) Using Hoeffding's inequality, $\mathbb{P}(\bigcup_{n=1}^K F_n^r) \leq |\mathcal{S}||\mathcal{A}|HK \frac{\delta}{16|\mathcal{S}|^2|\mathcal{A}|HK} \leq \frac{\delta'}{|\mathcal{S}|}$.

(ii) Using Hoeffding's inequality, $\mathbb{P}(\bigcup_{n=1}^K F_n^c) \leq |\mathcal{S}||\mathcal{A}|HK \frac{\delta}{16|\mathcal{S}|^2|\mathcal{A}|HK} \leq \frac{\delta'}{|\mathcal{S}|}$.

(iii) Using Hoeffding's inequality, $\mathbb{P}(\bigcup_{n=1}^K F_n^P) \leq |\mathcal{S}|^2|\mathcal{A}|HK \frac{\delta}{16|\mathcal{S}|^2|\mathcal{A}|HK} \leq \delta'$.

(iv) Using Bernstein's inequality, $\mathbb{P}(\bigcup_{n=1}^K \tilde{F}_n^P) \leq |\mathcal{S}|^2|\mathcal{A}|HK \frac{\delta}{16|\mathcal{S}|^2|\mathcal{A}|HK} \leq \delta'$.

Using union bound, $\mathbb{P}(\bigcup_{n=1}^K F_n^r \cup F_n^c \cup F_n^P \cup \tilde{F}_n^P) \leq \mathbb{P}(\bigcup_{n=1}^K F_n^r) + \mathbb{P}(\bigcup_{n=1}^K F_n^c) + \mathbb{P}(\bigcup_{n=1}^K F_n^P) + \mathbb{P}(\bigcup_{n=1}^K \tilde{F}_n^P) \leq (2 + 2/|\mathcal{S}|)\delta' = \delta/2$.

The episodic CMDP problem studied here can be simulated as follows: at episode $k$, taking action $a$ at state $s$ and step $h$ returns

$$\left( R^{n_k(s,a,h)}(s,a,h), C^{n_k(s,a,h)}(s,a,h), S^{n_k(s,a,h)}(s,a,h) \right).$$

Then all the samples drawn in the episodic CMDP problem are contained in

$$\{R^n(s,a,h), C^n(s,a,h), S^n(s,a,h)\}_{n=1}^K.$$

The sample averages calculated by the algorithm are

$$\left( \hat{r}_h^k(s,a), \hat{c}_h^k(s,a), \hat{P}_h^k(\cdot|s,a) \right) = \left( \hat{r}^{n_k(s,a,h)}(s,a,h), \hat{c}^{n_k(s,a,h)}(s,a,h), \hat{P}^{n_k(s,a,h)}(\cdot|s,a,h) \right).$$

Since $\mathcal{E}^{gen}$ implies $\mathcal{E}$, $\mathcal{E}$ holds with probability at least $1 - \delta$. $\qquad\square$

**Remark.** *The analysis in the rest of this appendix is conditioned on the good event $\mathcal{E}$. That is, if $(\Omega, \mathcal{F}, P)$ is the underlying probability space, then we suppose throughout the rest of the appendix that the sample point $\omega \in \mathcal{E}$.*

## B  Details of analysis for the zero constraint violation case

### B.1  Constraint violation analysis

We will provide a proof of a slightly stronger result. Specifically, we will suppose that we know a policy $\pi^0$ and a constant $c^0 < \tau$ such that the safety cost $V_1^{\pi^0}(\mu; c, P) \leq c^0$. The strengthening lies

in the relaxed requirement of knowing only an *upper bound* on the safety cost of $\pi^0$, rather than its *exact* value.

The zero constraint violation of OptPess-LP is guaranteed by the pessimistic policy set $\Pi^k$, as stated in the following lemma. Recall

$$\Pi^k := \begin{cases} \{\pi^0\} & \text{if } V_1^{\pi^0}(\mu; \underline{c}^k, \hat{P}^k) \geq (\tau + c^0)/2, \\ \{\pi : V_1^\pi(\mu; \underline{c}^k, \hat{P}^k) \leq \tau\} & \text{otherwise.} \end{cases}$$

**Lemma B.1** (Restatement of Lemma 5.1). *On the good event $\mathcal{E}$, for any $k \in [K]$ and policy $\pi \in \Pi^k$, $V_1^\pi(\mu; c, P) \leq \tau$.*

*Proof of Lemma 5.1.* Fix $\omega \in \mathcal{E}$ throughput. Then, for any $k, s, a, h$, we have

$$\begin{aligned} (\underline{c}_h^k - c_h)(s, a) &= \underline{c}_h^k(s, a) - \hat{c}_h^k(s, a) + \hat{c}_h^k(s, a) - c_h(s, a) \\ &= (1 + |\mathcal{S}|H)\beta_h^k(s, a) + \hat{c}_h^k(s, a) - c_h(s, a) \\ &\geq (1 + |\mathcal{S}|H)\beta_h^k(s, a) - \beta_h^k(s, a) = |\mathcal{S}|H\beta_h^k(s, a), \end{aligned}$$

and

$$\sum_{s'} (\hat{P}_h^k - P_h)(s'|s, a)V_{h+1}^\pi(s'; c, P) \geq -H\sum_{s'} \beta_h^k(s, a) = -|\mathcal{S}|H\beta_h^k(s, a).$$

Thus, by Lemma D.2, for any policy $\pi$, we have

$$V_1^\pi(\mu; \underline{c}^k, \hat{P}^k) - V_1^\pi(\mu; c, P)$$

$$= \mathbb{E}_{\mu, \hat{P}^k, \pi}\left[ \sum_{h=1}^H \left( (\underline{c}_h^k - c_h)(S_h, A_h) + \sum_{s'} (\hat{P}_h^k - P_h)(s'|S_h, A_h)V_{h+1}^\pi(s'; c, P) \right) \Big| \mathcal{F}_{k-1} \right]$$

$$\geq \mathbb{E}_{\mu, \hat{P}^k, \pi}\left[ \sum_{h=1}^H |\mathcal{S}|H\beta_h^k(S_h, A_h) - |\mathcal{S}|H\beta_h^k(S_h, A_h) \Big| \mathcal{F}_{k-1} \right] \geq 0. \tag{17}$$

Consider any $\pi \in \Pi^k$. If $\pi = \pi^0$, then $V_1^{\pi^0}(\mu; c, P) = c^0 < \tau$. Otherwise,

$$V_1^\pi(\mu; c, P) \leq V_1^\pi(\mu; \underline{c}^k, \hat{P}^k) \leq \tau.$$

$\square$

## B.2 Regret of reward analysis

When the parameters $c, P$ are not well estimated, there may not exist any policy $\pi$ such that $V_1^\pi(\mu; \underline{c}^k, \hat{P}^k) \leq \tau$. Hence, as defined in (B.1), $\Pi^k$ is a singleton set $\{\pi^0\}$, and accordingly $\pi^0$ is executed, even though $\pi^0$ is not safe for $(\underline{c}^k, \hat{P}^k)$. It subsequently takes several episodes of exploration using $\pi^0$ until it becomes strictly safe for $(\underline{c}^k, \hat{P}^k)$, which means that $\pi^k = \pi^0$ and $V_1^{\pi^k}(\mu; \underline{c}^k, \hat{P}^k) \geq (\tau + c^0)/2$ when $|\Pi^k| = 1$. At that time, policies close enough to $\pi^0$, of which there are infinitely many, are also safe for $(\underline{c}^k, \hat{P}^k)$, and so $|\Pi^k| = +\infty$. At this point the learning algorithm can proceed to enhance reward while maintaining safety with respect to $(\underline{c}^k, \hat{P}^k)$.

To analyze the algorithm, we decompose the reward regret as follows:

$$Reg(K; r) = \underbrace{\sum_{k=1}^K \mathbb{1}(|\Pi^k| = 1)\left( V_1^{\pi^*}(\mu; r, P) - V_1^{\pi^0}(\mu; r, P) \right)}_{\text{(I)}}$$

$$+ \underbrace{\sum_{k=1}^K \mathbb{1}(|\Pi^k| > 1)\left( V_1^{\pi^*}(\mu; r, P) - V_1^{\pi^k}(\mu; \tilde{r}^k, \hat{P}^k) \right)}_{\text{(II)}}$$

$$+ \underbrace{\sum_{k=1}^K \mathbb{1}(|\Pi^k| > 1)\left( V_1^{\pi^k}(\mu; \tilde{r}^k, \hat{P}^k) - V_1^{\pi^k}(\mu; r, P) \right)}_{\text{(III)}}.$$

We analyze these three terms, and the results are summarized in the following lemmas, respectively.

The following lemma gives an upper bound on the number of episodes for exploration by policy $\pi^0$. It thereby bounds the first term on the right-hand side (RHS) of (13).

**Lemma B.2** (Restatement of Lemma 5.2). *On the good event $\mathcal{E}$, $\sum_{k=1}^{K} \mathbb{1}(|\Pi^k| = 1) \leq C'$, where $C' = \tilde{\mathcal{O}}(H^4|\mathcal{S}|^3|\mathcal{A}|/((\tau - c^0)^2 \wedge (\tau - c^0)))$.*

*Proof of Lemma 5.2.* With $K' := \sum_{k=1}^{K} \mathbb{1}(|\Pi^k| = 1)$, we then have

$$\frac{(\tau - c^0)K'}{2} = \sum_{k=1}^{K} \mathbb{1}(|\Pi^k| = 1)\frac{(\tau - c^0)}{2}$$

$$= \sum_{k=1}^{K} \mathbb{1}(|\Pi^k| = 1)\left(\frac{\tau + c^0}{2} - c^0\right)$$

$$\overset{(a)}{\leq} \sum_{k=1}^{K} \mathbb{1}(V_1^{\pi^k}(\mu; \underline{c}^k, \hat{P}^k) \geq \frac{\tau + c^0}{2})\left(V_1^{\pi^k}(\mu; \underline{c}^k, \hat{P}^k) - V_1^{\pi^k}(\mu; c, P)\right)$$

$$\overset{(b)}{=} \tilde{\mathcal{O}}\left(H^4|\mathcal{S}|^3|\mathcal{A}| + H^2\sqrt{|\mathcal{S}|^3|\mathcal{A}|K'}\right).$$

(a) holds since $\pi^k = \pi^0$, $V_1^{\pi^k}(\mu; \underline{c}^k, \hat{P}^k) \geq (\tau + c^0)/2$, and $V_1^{\pi^k}(\mu; c, P) \leq c^0$ when $|\Pi^k| = 1$. The equality (b) follows from Lemma D.4 with $|\underline{c}_h^k - c_h| = |\hat{c}_h^k - c_h + (1 + H|\mathcal{S}|)\beta_h^k| \leq (2 + |\mathcal{S}|H)\beta_h^k$. Applying Lemma D.6 for $K'$, there exists some parameter $C'$ that

$$K' \leq C' = \tilde{\mathcal{O}}\left(\frac{H^4|\mathcal{S}|^3|\mathcal{A}|}{(\tau - c^0)(1 \wedge (\tau - c^0))}\right).$$

$\square$

The following lemma gives the resulting high probability bound for Term (II).

**Lemma B.3** (Restatement of Lemma 5.3). *Recall $\alpha_r = 1 + |\mathcal{S}|H + 4H(1 + |\mathcal{S}|H)/(\tau - c^0)$. On the good event $\mathcal{E}$,*

$$\sum_{k=1}^{K} \mathbb{1}(|\Pi^k| > 1)\left(V_1^{\pi^*}(\mu; r, P) - V_1^{\pi^k}(\mu; \bar{r}^k, \hat{P}^k)\right) \leq 0.$$

*Proof of Lemma 5.3.* Consider any $k \in [K]$ with $V_1^{\pi^0}(\mu; \underline{c}^k, \hat{P}^k) < (\tau + c^0)/2$. Then $|\Pi^k| > 1$. When $\pi^* \in \Pi^k$, it is straightforward to verify that Term (II) is less or equal to zero by optimism. When $\pi^* \notin \Pi^k$, define a probabilistic mixed policy

$$\tilde{\pi}^k = B_{\gamma_k}\pi^* + (1 - B_{\gamma_k})\pi^0,$$

where $B_{\gamma_k}$ is an independent Bernoulli distributed random variable with mean $\gamma_k$. Let $\gamma_k \in [0, 1]$ be the largest coefficient such that

$$V_1^{\tilde{\pi}^k}(\mu; \underline{c}^k, \hat{P}^k) \leq \tau. \tag{18}$$

If $V_1^{\pi^*}(\mu; \underline{c}^k, \hat{P}^k) < \tau$, then $\gamma_k = 1$. Otherwise, at $\gamma_k$, equality holds in (18). Then we have

$$\tau = \gamma_k V_1^{\pi^*}(\mu; \underline{c}^k, \hat{P}^k) + (1 - \gamma_k)V_1^{\pi^0}(\mu; \underline{c}^k, \hat{P}^k)$$

$$\leq \gamma_k V_1^{\pi^*}(\mu; \underline{c}^k, \hat{P}^k) + (1 - \gamma_k)\frac{\tau + c^0}{2}$$

$$= \gamma_k(V_1^{\pi^*}(\mu; \underline{c}^k, \hat{P}^k) - V_1^{\pi^*}(\mu; c, P)) + \gamma_k V_1^{\pi^*}(\mu; c, P) + (1 - \gamma_k)\frac{\tau + c^0}{2}$$

$$\leq \gamma_k\left(V_1^{\pi^*}(\mu; \underline{c}^k, \hat{P}^k) - V_1^{\pi^*}(\mu; c, P)\right) + \gamma_k\tau + \frac{\tau + c^0}{2} - \gamma_k\frac{\tau + c^0}{2}$$

$$= \gamma_k\left(V_1^{\pi^*}(\mu; \underline{c}^k, \hat{P}^k) - V_1^{\pi^*}(\mu; c, P) + \frac{\tau - c^0}{2}\right) + \frac{\tau + c^0}{2}.$$

From (17), $V_1^{\pi^*}(\mu;\underline{c}^k,\hat{P}^k) - V_1^{\pi^*}(\mu;c,P) + (\tau - c^0)/2 > 0$, from which it follows that

$$\gamma_k \geq \frac{\tau - c^0}{\tau - c^0 + 2\left(V_1^{\pi^*}(\mu;\underline{c}^k,\hat{P}^k) - V_1^{\pi^*}(\mu;c,P)\right)}.$$

By Lemma D.2, for any policy $\pi$,

$$V_1^{\pi}(\mu;\underline{c}^k,\hat{P}^k) - V_1^{\pi}(\mu;c,P) = \mathbb{E}_{\mu,\hat{P}^k,\pi}\left[\sum_{h=1}^{H}(\underline{c}_h^k - c_h)(S_h,A_h)\,\Big|\,\mathcal{F}_{k-1}\right]$$

$$+ \mathbb{E}_{\mu,\hat{P}^k,\pi}\left[\sum_{h=1}^{H}\langle(\hat{P}_h^k - P_h)(\cdot|S_h,A_h),V_{h+1}^{\pi}(\cdot;c,P)\rangle\,\Big|\,\mathcal{F}_{k-1}\right]$$

$$\overset{(a)}{\leq} \mathbb{E}_{\mu,\hat{P}^k,\pi}\left[\sum_{h=1}^{H}2(1+|\mathcal{S}|H)\beta_h^k(S_h,A_h))\,\Big|\,\mathcal{F}_{k-1}\right]$$

$$= 2(1+|\mathcal{S}|H)V_1^{\pi}(\mu;\beta^k,\hat{P}^k),$$

where (a) holds since for any $k,s,a,h$, we have

$$(\underline{c}_h^k - c_h)(s,a) = \underline{c}_h^k(s,a) - \hat{c}_h^k(s,a) + \hat{c}_h^k(s,a) - c_h(s,a)$$

$$\leq (1+|\mathcal{S}|H)\beta_h^k(s,a) + \beta_h^k(s,a)$$

$$= (2+|\mathcal{S}|H)\beta_h^k(s,a),$$

and

$$\sum_{s'}(\hat{P}_h^k - P_h)(s'|s,a)V_{h+1}^{\pi}(s';c,P) \leq H\sum_{s'}\beta_h^k(s,a) = |\mathcal{S}|H\beta_h^k(s,a).$$

Though policy $\tilde{\pi}^k$ is not a (randomized or non-randomized) Markov policy, we can find a randomized Markov policy $\hat{\pi}^k$ such that the occupation distributions of the state-action pair at any time of $\hat{\pi}^k$ coincides with $\tilde{\pi}^k$ under the transition probabilities $\hat{P}^k$. Therefore, as stated in by Lemma D.3, $V_1^{\hat{\pi}^k}(\mu;g,\hat{P}^k) = V_1^{\tilde{\pi}^k}(\mu;g,\hat{P}^k)$ for any $g$.

Since $\hat{\pi}^k \in \Pi^k$, by the definition of $\pi^k$, we then have

$$V_1^{\pi^k}(\mu;\bar{r}^k,\hat{P}^k) \geq V_1^{\hat{\pi}^k}(\mu;\bar{r}^k,\hat{P}^k) = V_1^{\tilde{\pi}^k}(\mu;\bar{r}^k,\hat{P}^k)$$

$$= \gamma_k V_1^{\pi^*}(\mu;\bar{r}^k,\hat{P}^k) + (1-\gamma_k)V_1^{\pi^0}(\mu;\bar{r}^k,\hat{P}^k)$$

$$\geq \gamma_k V_1^{\pi^*}(\mu;\bar{r}^k,\hat{P}^k)$$

$$= \frac{\tau - c^0}{\tau - c^0 + 2\left(V_1^{\pi^*}(\mu;\underline{c}^k,\hat{P}^k) - V_1^{\pi^*}(\mu;c,P)\right)}V_1^{\pi^*}(\mu;\bar{r}^k,\hat{P}^k)$$

$$\geq \frac{\tau - c^0}{\tau - c^0 + 4(1+|\mathcal{S}|H)V_1^{\pi^*}(\mu;\beta^k,\hat{P}^k)}V_1^{\pi^*}(\mu;\bar{r}^k,\hat{P}^k).$$

To make $V_1^{\pi^k}(\mu;\bar{r}^k,\hat{P}^k) \geq V_1^{\pi^*}(\mu;r,P)$, it suffices that

$$\frac{\tau - c^0}{\tau - c^0 + 4(1+|\mathcal{S}|H)V_1^{\pi^*}(\mu;\beta^k,\hat{P}^k)}V_1^{\pi^*}(\mu;\bar{r}^k,\hat{P}^k) \geq V_1^{\pi^*}(\mu;r,P),$$

which follows since

$$(\tau - c^0)(V_1^{\pi^*}(\mu;\bar{r}^k,\hat{P}^k) - V_1^{\pi^*}(\mu;r,P)) \geq 4(1+|\mathcal{S}|H)V_1^{\pi^*}(\mu;\beta^k,\hat{P}^k)V_1^{\pi^*}(\mu;r,P).$$

Now notice that from Lemma D.2,

$$V_1^{\pi^*}(\mu; \bar{r}^k, \hat{P}^k) - V_1^{\pi^*}(\mu; r, P)$$

$$= \mathbb{E}_{\mu, \hat{P}^k, \pi^*} \left[ \sum_{h=1}^{H} \left( (\bar{r}_h^k - r_h)(S_h, A_h) + \sum_{s'} (\hat{P}_h^k - P_h)(s'|S_h, A_h)V_{h+1}^{\pi^*}(s'; r, P) \right) \Big| \mathcal{F}_{k-1} \right]$$

$$\geq \mathbb{E}_{\mu, \hat{P}^k, \pi^*} \left[ \sum_{h=1}^{H} (\alpha_r - 1 - H|\mathcal{S}|)\beta_h^k(S_h, A_h) \Big| \mathcal{F}_{k-1} \right]$$

$$= (\alpha_r - 1 - H|\mathcal{S}|)V_1^{\pi^*}(\mu; \beta^k, \hat{P}^k).$$

Thus, the choice of $\alpha_r = 4H(1 + |\mathcal{S}|H)/(\tau - c^0) + 1 + |\mathcal{S}|H$ suffices to guarantee that $V_1^{\pi^k}(\mu; \bar{r}^k, \hat{P}^k) \geq V_1^{\pi^*}(\mu; r, P)$ for any $k$ with $|\Pi^k| > 1$. $\qquad \square$

Finally, Term (III) is bounded as follows by employing the value of $\alpha_r$.

**Lemma B.4** (Restatement of Lemma 5.4). *On the good event $\mathcal{E}$,*

$$\sum_{k=1}^{K} \mathbb{1}(|\Pi^k| > 1)\left(V_1^{\pi^k}(\mu; \bar{r}^k, \hat{P}^k) - V_1^{\pi^k}(\mu; r, P)\right) = \tilde{\mathcal{O}}\left(\frac{H^3}{\tau - c^0}\sqrt{|\mathcal{S}|^3|\mathcal{A}|K} + \frac{H^5|\mathcal{S}|^3|\mathcal{A}|}{\tau - c^0}\right).$$

*Proof of Lemma 5.4.* Since $|\bar{r}_h^k - r_h| = |\hat{r}_h^k - r_h + \alpha_r \beta_h^k| \leq (1 + \alpha_r)\beta_h^k$, by Lemma D.4, we have

$$\sum_{k=1}^{K} \mathbb{1}(|\Pi^k| > 1)\left(V_1^{\pi^k}(\mu; \bar{r}^k, \hat{P}^k) - V_1^{\pi^k}(\mu; r, P)\right)$$

$$\leq \sum_{k=1}^{K} \left|V_1^{\pi^k}(\mu; \bar{r}^k, \hat{P}^k) - V_1^{\pi^k}(\mu; r, P)\right|$$

$$= \tilde{\mathcal{O}}\left((\alpha_r + H\sqrt{|\mathcal{S}|})H\sqrt{|\mathcal{S}||\mathcal{A}|K} + H^3|\mathcal{S}|^2|\mathcal{A}|(\alpha_r + |\mathcal{S}|)\right)$$

$$= \tilde{\mathcal{O}}\left(\frac{H^3}{\tau - c^0}\sqrt{|\mathcal{S}|^3|\mathcal{A}|K} + \frac{H^5|\mathcal{S}|^3|\mathcal{A}|}{\tau - c^0}\right).$$

$\qquad \square$

## B.3   Proof of Theorem 3.1

**Theorem B.5** (Regret and constraint violation bounds for OptPess-LP (Restatement of Theorem 3.1)). *On the good event $\mathcal{E}$, OptPess-LP has zero constraint violation with*

$$Reg^{\textbf{\textit{OPLP}}}(K; r) = \tilde{\mathcal{O}}\left(\frac{H^3}{\tau - c^0}\sqrt{|\mathcal{S}|^3|\mathcal{A}|K} + \frac{H^5|\mathcal{S}|^3|\mathcal{A}|}{(\tau - c^0)^2 \wedge (\tau - c^0)}\right), \textit{ where } a \wedge b := \min\{a, b\}.$$

*Proof of Theorem 3.1.* The zero constraint violation of OptPess-LP is guaranteed by Lemma 5.1. Combining Lemmas 5.2, 5.3, and 5.4, which give upper bounds to Terms (I)-(III) respectively, we have

$$\sum_{k=1}^{K} \left(V_1^{\pi^*}(\mu; r, P) - V_1^{\pi^k}(\mu; r, P)\right) = \tilde{\mathcal{O}}\left(\frac{H^3}{\tau - c^0}\sqrt{|\mathcal{S}|^3|\mathcal{A}|K} + \frac{H^5|\mathcal{S}|^3|\mathcal{A}|}{(\tau - c^0) \wedge (\tau - c^0)^2}\right).$$

$\qquad \square$

# C  Details of analysis for bounded constraint violation case

Recall $\epsilon_k = 5H^2\sqrt{|\mathcal{S}|^3|\mathcal{A}|}(\log\frac{k}{\delta'}+1)/\sqrt{k\log\frac{k}{\delta'}}$, where $\delta' = \delta/(16|\mathcal{S}|^2|\mathcal{A}|H)$. The existence of a feasible solution to (10) can be guaranteed if $\epsilon_k \leq \tau - c^0$. Let $C''$ be the smallest value such that $\forall k \geq C''$, $\epsilon_k \leq (\tau - c^0)/2$. Then the perturbed optimization problem (10) has at least one feasible solution for any $k \geq C''$. By simple calculation, one can verify that $C'' = \mathcal{O}(\frac{H^4|\mathcal{S}|^3|\mathcal{A}|}{(\tau-c^0)^2}\log\frac{H^4|\mathcal{S}|^3|\mathcal{A}|}{(\tau-c^0)^2\delta'})$. Notice that since $\epsilon_k$ is a function not depending on $K$, so is the coefficient $C''$.

**Lemma C.1** (Optimistic of cost value). *On the good event $\mathcal{E}$, for any $C'' \leq k \leq K$ and any policy $\pi$,*

$$\hat{V}_1^\pi(\mu; \tilde{c}^k, \hat{P}^k) \leq V_1^\pi(\mu; c, P).$$

*Proof of Lemma C.1.* If $\hat{V}_1^\pi(\mu; \tilde{c}^k, \hat{P}^k) = V_1^\pi(\mu; \tilde{c}^k, \hat{P}^k)$, then

$$\sum_{k=C''}^K \left(\hat{V}_1^\pi(\mu; \tilde{c}^k, \hat{P}^k) - V_1^\pi(\mu; c, P)\right)$$

$$= \sum_{k=C''}^K \mathbb{E}_{\mu,\hat{P}^k,\pi}\left[\sum_{h=1}^H \left(\tilde{c}_h^k(S_h, A_h) - c_h(S_h, A_h)\right) + \langle(\hat{P}_h^k - P_h)(\cdot|S_h, A_h), V_{h+1}^\pi(\cdot; c, P)\rangle \,\Big|\, \mathcal{F}_{k-1}\right]$$

$$\leq \sum_{k=C''}^K \mathbb{E}_{\mu,\hat{P}^k,\pi}\left[\sum_{h=1}^H (\hat{c}_h^k(S_h, A_h) - c_h(S_h, A_h)) - \beta_h^k - H|\mathcal{S}|\beta_h^k + H|\mathcal{S}|\beta_h^k \,\Big|\, \mathcal{F}_{k-1}\right] \leq 0.$$

Otherwise, $\hat{V}_1^\pi(\mu; \tilde{c}^k, \hat{P}^k) = 0 \leq V_1^\pi(\mu; c, P)$. $\qquad\square$

## C.1  Constraint violation analysis

**Lemma C.2** (Restatement of Lemma 5.5). *Recall $\delta' = \delta/(16|\mathcal{S}|^2|\mathcal{A}|H)$. On the good event $\mathcal{E}$,*

$$\sum_{k=1}^K \left(V_1^{\pi^k}(\mu; c, P) - \hat{V}_1^{\pi^k}(\mu; \tilde{c}^k, \hat{P}^k)\right) \leq 8H^2\sqrt{|\mathcal{S}|^3|\mathcal{A}|K\log\frac{K}{\delta'}} + \tilde{\mathcal{O}}\left(H^4|\mathcal{S}|^3|\mathcal{A}|\right),$$

*where $\tilde{\mathcal{O}}(H^4|\mathcal{S}|^3|\mathcal{A}|)$ contains factors poly-logarithmic in $K$.*

*Proof of Lemma 5.5.* For any $k, s, a, h$, we have

$$|(\tilde{c}_h^k - c_h)(s, a)| = |\tilde{c}_h^k(s, a) - \hat{c}_h^k(s, a) + \hat{c}_h^k(s, a) - c_h(s, a)|$$
$$\leq (1 + |\mathcal{S}|H)\beta_h^k(s, a) + |\hat{c}_h^k(s, a) - c_h(s, a)|$$
$$\leq (1 + |\mathcal{S}|H)\beta_h^k(s, a) + \beta_h^k(s, a) = (2 + |\mathcal{S}|H)\beta_h^k(s, a).$$

By Lemma D.4, we have

$$\sum_{k=1}^K \left(V_1^{\pi^k}(\mu; c, P) - \hat{V}_1^{\pi^k}(\mu; \tilde{c}^k, \hat{P}^k)\right)$$

$$\leq (3(2 + |\mathcal{S}|H) + 3\sqrt{2}H\sqrt{|\mathcal{S}|})H\sqrt{|\mathcal{S}||\mathcal{A}|KZ} + \tilde{\mathcal{O}}\left(H^4|\mathcal{S}|^3|\mathcal{A}|\right)$$

$$= (6 + 3|\mathcal{S}|H + 3H\sqrt{2|\mathcal{S}|})H\sqrt{|\mathcal{S}||\mathcal{A}|KZ} + \tilde{\mathcal{O}}\left(H^4|\mathcal{S}|^3|\mathcal{A}|\right)$$

$$\overset{(a)}{\leq} 8H^2|\mathcal{S}|\sqrt{|\mathcal{S}||\mathcal{A}|KZ} + \tilde{\mathcal{O}}\left(H^4|\mathcal{S}|^3|\mathcal{A}|\right),$$

where $\tilde{\mathcal{O}}(H^4|\mathcal{S}|^3|\mathcal{A}|)$ contains factors poly-logarithmic in $K$. (a) holds since we assume that $|\mathcal{S}| \geq 2$ and $H \geq 2$. $\qquad\square$

**Lemma C.3** (Restatement of Lemma 5.6). *On the good event $\mathcal{E}$, for any $k \geq C''$,*

$$\lambda^k \leq \frac{1}{\zeta}\ln\frac{11\nu_{\max}^2}{3\rho^2} + C''(H - \tau) + \sum_{u=1}^{C''}\epsilon_u + H + \frac{4(H^2 + \epsilon_k^2 + \eta^k H)}{\tau - c^0},$$

*where $\rho := (\tau - c^0)/4$, $\nu_{\max} := H$, $\zeta := \rho/(\nu_{\max}^2 + \nu_{\max}\rho/3)$.*

*Proof of Lemma 5.6.* We will first verify conditions (i) and (ii) of Lemma D.7 for $k \geq C''$ by defining $\Phi(k) = \lambda^k$. Suppose $\Phi(k) = \lambda^k \geq \varphi_k := 4(H^2 + \epsilon_k^2 + \eta^k H)/(\tau - c^0)$. It is straightforward to verify that $\lambda^k + \hat{V}_1^{\pi^k}(\mu; \tilde{c}^k, \hat{P}^k) + \epsilon_k - \tau$ is positive, which implies

$$\lambda^{k+1} = \lambda^k + \hat{V}_1^{\pi^k}(\mu; \tilde{c}^k, \hat{P}^k) + \epsilon_k - \tau.$$

Let $L(k) = (\lambda^k)^2/2$, we have

$$
\begin{aligned}
L(k+1) - L(k) &= \frac{1}{2}(\lambda^{k+1})^2 - \frac{1}{2}(\lambda^k)^2 \\
&= \lambda^k(\lambda^{k+1} - \lambda^k) + \frac{1}{2}(\lambda^{k+1} - \lambda^k)^2 \\
&= \lambda^k\left(\hat{V}_1^{\pi^k}(\mu; \tilde{c}^k, \hat{P}^k) + \epsilon_k - \tau\right) + \frac{1}{2}(\hat{V}_1^{\pi^k}(\mu; \tilde{c}^k, \hat{P}^k) + \epsilon_k - \tau)^2 \\
&= \left(\lambda^k\left(\hat{V}_1^{\pi^k}(\mu; \tilde{c}^k, \hat{P}^k) + \epsilon_k - \tau\right) - \eta^k\hat{V}_1^{\pi^k}(\mu; \tilde{r}^k, \hat{P}^k)\right) \\
&\quad + \eta^k\hat{V}_1^{\pi^k}(\mu; \tilde{r}^k, \hat{P}^k) + \frac{1}{2}(\hat{V}_1^{\pi^k}(\mu; \tilde{c}^k, \hat{P}^k) + \epsilon_k - \tau)^2 \\
&\stackrel{(a)}{\leq} \left(\lambda^k\left(\hat{V}_1^{\pi^0}(\mu; \tilde{c}^k, \hat{P}^k) + \epsilon_k - \tau\right) - \eta^k\hat{V}_1^{\pi^0}(\mu; \tilde{r}, \hat{P}^k)\right) \\
&\quad + \eta^k\hat{V}_1^{\pi^k}(\mu; \tilde{r}^k, \hat{P}^k) + (\hat{V}_1^{\pi^k}(\mu; \tilde{c}^k, \hat{P}^k) - \tau)^2 + \epsilon_k^2 \\
&\stackrel{(b)}{\leq} \lambda^k\left(V_1^{\pi^0}(\mu; c, P) + \epsilon_k - \tau\right) + \eta^k H + H^2 + \epsilon_k^2 \\
&\leq -\frac{\tau - c^0}{2}\lambda^k + H^2 + \epsilon_k^2 + \eta^k H.
\end{aligned}
$$

(a) is by the optimality of $\pi^k$ and $(a+b)^2/2 \leq (a^2+b^2)$. (b) holds since $\hat{V}_1^{\pi^0}(\mu; \tilde{c}^k, \hat{P}^k) \leq V_1^{\pi^0}(\mu; c, P)$ according to Lemma C.1.

It then follows that $L(k+1) < L(k)$ and

$$
\begin{aligned}
\lambda^{k+1} - \lambda^k &= \frac{(\lambda^{k+1})^2 - (\lambda^k)^2}{\lambda^{k+1} + \lambda^k} \\
&\leq \frac{(\lambda^{k+1})^2 - (\lambda^k)^2}{2\lambda^k} \\
&\leq -\frac{\tau - c^0}{2} + \frac{H^2 + \epsilon_k^2 + \eta^k H}{\varphi_k} = -\frac{\tau - c^0}{4} =: -\rho.
\end{aligned}
$$

In addition, since $\forall k \geq C''$, $\epsilon_k \leq (\tau - c^0)/2$,

$$|\lambda^{k+1} - \lambda^k| \leq |\hat{V}_1^{\pi^k}(\mu; \tilde{c}^k, \hat{P}^k) + \epsilon_k - \tau| \leq H =: \nu_{\max}.$$

With the definitions of $\varphi_k, \rho, \nu_{\max}$, we apply Lemma D.7 starting from $C''$, which gives

$$
\begin{aligned}
\lambda^k &\leq \frac{1}{\zeta}\ln\left(e^{\zeta\lambda^{C''}} + \frac{2e^{\zeta(v_{\max}+\varphi_k)}}{\zeta\rho}\right) \\
&\leq \frac{1}{\zeta}\ln\left(\frac{11v_{\max}^2 e^{\zeta(\lambda^{C''}+v_{\max}+\varphi_k)}}{3\rho^2}\right) \\
&= \frac{1}{\zeta}\ln\frac{11\nu_{\max}^2}{3\rho^2} + \lambda^{C''} + v_{\max} + \varphi_k \\
&\leq \frac{1}{\zeta}\ln\frac{11\nu_{\max}^2}{3\rho^2} + C''(H - \tau) + \sum_{u=1}^{C''}\epsilon_u + H + \frac{4(H^2 + \epsilon_k^2 + \eta^k H)}{\tau - c^0},
\end{aligned}
$$

where $\zeta = \rho/(\nu_{\max}^2 + \nu_{\max}\rho/3) \geq 3(\tau - c^0)/(13H^2)$, and the last inequality is by

$$\lambda^{C''} \leq \lambda^1 + \sum_{k=1}^{C''-1} (\hat{V}_1^{\pi^k}(\mu; \tilde{c}^k, \hat{P}^k) + \epsilon_k - \tau)_+$$

$$\leq \sum_{k=1}^{C''} \epsilon_k + C''(H - \tau). \tag{19}$$

$\square$

The bounded constraint violation of OptPess-PrimalDual is established as follows. Notice that

$$\lambda^{K+1} = \left(\lambda^K + \hat{V}_1^{\pi^K}(\mu; \tilde{c}^K, \hat{P}^K) + \epsilon_K - \tau\right)_+$$

$$\geq \epsilon_K + \hat{V}_1^{\pi^K}(\mu; \tilde{c}^K, \hat{P}^K) - \tau + \lambda^K$$

$$\geq \epsilon_K + \epsilon_{K-1} + \hat{V}_1^{\pi^K}(\mu; \tilde{c}^K, \hat{P}^K) - \tau + \hat{V}_1^{\pi^{K-1}}(\mu; \tilde{c}^{K-1}, \hat{P}^{K-1}) - \tau + \lambda^{K-1}$$

$$\geq \cdots$$

$$\geq \sum_{k=1}^K \epsilon_k + \sum_{k=1}^K \left(\hat{V}_1^{\pi^k}(\mu; \tilde{c}^k, \hat{P}^k) - \tau\right) + \lambda^1.$$

Since $\lambda^1 = 0$ according to the initial condition of Algorithm 2, we can decompose the constraint violation:

$$Reg^{\mathbf{OPPD}}(K; c) = \left(\sum_{k=1}^K \left(V_1^{\pi^k}(\mu; c, P) - \hat{V}_1^{\pi^k}(\mu; \tilde{c}^k, \hat{P}^k)\right) + \sum_{k=1}^K \left(\hat{V}_1^{\pi^k}(\mu; \tilde{c}^k, \hat{P}^k) - \tau\right)\right)_+$$

$$\leq \left(\sum_{k=1}^K \left(V_1^{\pi^k}(\mu; c, P) - \hat{V}_1^{\pi^k}(\mu; \tilde{c}^k, \hat{P}^k)\right) + \lambda^{K+1} - \sum_{k=1}^K \epsilon_k\right)_+.$$

The first summation term can be bounded by Lemma 5.5. The second $\lambda^{K+1}$ term can be bounded by Lemma 5.6. Finally, recall $\epsilon_k = 5H^2\sqrt{|\mathcal{S}|^3|\mathcal{A}|}(\log\frac{k}{\delta'} + 1)/\sqrt{k\log\frac{k}{\delta'}}$, we have

$$\sum_{k=1}^K \epsilon_k \geq \int_1^{K+1} \epsilon_u du$$

$$\geq 10H^2\sqrt{|\mathcal{S}|^3|\mathcal{A}|K\log\frac{K}{\delta'}} - 10H^2\sqrt{|\mathcal{S}|^3|\mathcal{A}|\log\frac{1}{\delta'}}.$$

The negative $\sum_{k=1}^K \epsilon_k$ term can cancel the dominant terms. Thus, with probability at least $(1 - \delta)$, the violation is bounded as

$$Reg^{\mathbf{OPPD}}(K; c) = \mathcal{O}\left(C''H + H^2\sqrt{|\mathcal{S}|^3|\mathcal{A}|C''\log(C''/\delta')}\right).$$

## C.2 Regret of reward analysis

For episode $k$ with $k \geq C''$, let $\pi^{\epsilon_k,*}$ be the optimal policy for (10), which is well-defined by the definition of $C''$. We decompose the reward regret as

$$Reg^{\mathbf{OPPD}}(K; r) = \sum_{k=1}^{C''} \left(V_1^{\pi^*}(\mu; r, P) - V_1^{\pi^k}(\mu; r, P)\right)$$

$$+ \sum_{k=C''}^K \left(V_1^{\pi^*}(\mu; r, P) - V_1^{\pi^{\epsilon_k,*}}(\mu; r, P)\right) + \sum_{k=C''}^K \left(V_1^{\pi^{\epsilon_k,*}}(\mu; r, P) - \hat{V}_1^{\pi^{\epsilon_k,*}}(\mu; \tilde{r}^k, \hat{P}^k)\right)$$

$$+ \sum_{k=C''}^K \left(\hat{V}_1^{\pi^{\epsilon_k,*}}(\mu; \tilde{r}^k, \hat{P}^k) - \hat{V}_1^{\pi^k}(\mu; \tilde{r}^k, \hat{P}^k)\right) + \sum_{k=C''}^K \left(\hat{V}_1^{\pi^k}(\mu; \tilde{r}^k, \hat{P}^k) - V_1^{\pi^k}(\mu; r, P)\right).$$

$$\tag{20}$$

We upper bound each term in the RHS of (20). Since $V_1^\pi(\mu; r, P) \in [0, H]$ for any policy $\pi$, the first term is upper bounded by $HC''$. The second and third terms can be bounded by the following two lemmas.

**Lemma C.4** (Restatement of Lemma 5.7). *On the good event $\mathcal{E}$,*

$$\sum_{k=C''}^{K} \left( V_1^{\pi^*}(\mu; r, P) - V_1^{\pi^{\epsilon_k,*}}(\mu; r, P) \right) \leq \sum_{k=C''}^{K} \frac{\epsilon_k H}{\tau - c^0} = \tilde{\mathcal{O}} \left( \frac{H^3}{\tau - c^0} \sqrt{|\mathcal{S}|^3 |\mathcal{A}| K} \right).$$

*Proof of Lemma 5.7.* Define a probabilistic mixed policy

$$\pi^{\epsilon_k} = (1 - B_k)\pi^* + B_k \pi^0, \quad \forall k \geq C'',$$

where $B_k$ is a Bernoulli distributed random variable with mean $\epsilon_k / (\tau - c^0)$.

We have that

$$V_1^{\pi^{\epsilon_k}}(\mu; c, P) = (1 - \frac{\epsilon_k}{\tau - c^0}) V_1^{\pi^*}(\mu; c, P) + \frac{\epsilon_k}{\tau - c^0} V_1^{\pi^0}(\mu; c, P)$$

$$\leq (1 - \frac{\epsilon_k}{\tau - c^0})\tau + \frac{\epsilon_k}{\tau - c^0} c^0 = \tau - \epsilon_k.$$

Though $\pi^{\epsilon_k}$ is not a Markov policy, by Lemma D.3, there exists a Markov policy $\hat{\pi}^{\epsilon_k}$, which has the same performance as $\pi^{\epsilon_k}$ under transition probabilities $P$. $\hat{\pi}^{\epsilon_k}$ is a feasible solution to the following problem

$$\max_{\pi} \quad V_1^\pi(\mu; r, P)$$

$$\text{s.t.} \quad V_1^\pi(\mu; c, P) + \epsilon_k \leq \tau.$$

We then have

$$\sum_{k=C''}^{K} V_1^{\pi^*}(\mu; r, P) - V_1^{\pi^{\epsilon_k,*}}(\mu; r, P) \leq \sum_{k=C''}^{K} V_1^{\pi^*}(\mu; r, P) - V_1^{\hat{\pi}^{\epsilon_k}}(\mu; r, P)$$

$$= \sum_{k=C''}^{K} V_1^{\pi^*}(\mu; r, P) - V_1^{\pi^{\epsilon_k}}(\mu; r, P)$$

$$\leq \sum_{k=C''}^{K} \frac{\epsilon_k}{\tau - c^0} \left( V_1^{\pi^*}(\mu; r, P) - V_1^{\pi^0}(\mu; r, P) \right)$$

$$\leq \sum_{k=C''}^{K} \frac{\epsilon_k H}{\tau - c^0}.$$

$\square$

**Lemma C.5** (Restatement of Lemma 5.8). *On the good event $\mathcal{E}$,* $\sum_{k=C''}^{K} \left( V_1^{\pi^{\epsilon_k,*}}(\mu; r, P) - \hat{V}_1^{\pi^{\epsilon_k,*}}(\mu; \tilde{r}^k, \hat{P}^k) \right) \leq 0.$

*Proof of Lemma 5.8.* For any $k \in [C'', K]$, $\pi^{\epsilon_k,*}$ is well-defined. If $\hat{V}_1^{\pi^{\epsilon_k,*}}(\mu; \tilde{r}^k, \hat{P}^k) = V_1^{\pi^{\epsilon_k,*}}(\mu; \tilde{r}^k, \hat{P}^k)$, then by the value difference lemma (Lemma D.2), we know

$$V_1^{\pi^{\epsilon_k,*}}(\mu; r, P) - \hat{V}_1^{\pi^{\epsilon_k,*}}(\mu; \tilde{r}^k, \hat{P}^k)$$

$$= \mathbb{E}_{\mu, \hat{P}^k, \hat{\pi}^{\epsilon_k,*}} \left[ \sum_{h=1}^{H} \left( r_h(S_h, A_h) - \tilde{r}_h^k(S_h, A_h) + \sum_{s'} (P_h - \hat{P}_h^k)(s'|S_h, A_h) V_{h+1}^\pi(s'; r, P) \right) \Big| \mathcal{F}_{k-1} \right]$$

$$\leq \sum_{k=C''}^{K} \mathbb{E}_{\mu, \hat{P}^k, \hat{\pi}^{\epsilon_k,*}} \left[ \sum_{h=1}^{H} (r_h(S_h, A_h) - \hat{r}_h^k(S_h, A_h)) - \beta_h^k - H|\mathcal{S}|\beta_h^k + H|\mathcal{S}|\beta_h^k \Big| \mathcal{F}_{k-1} \right] \leq 0.$$

Otherwise, $\hat{V}_1^{\pi^{\epsilon_k,*}}(\mu; \tilde{r}^k, \hat{P}^k) = H \geq V_1^{\pi^{\epsilon_k,*}}(\mu; r, P)$. Thus $\hat{V}_1^{\pi^{\epsilon_k,*}}(\mu; \tilde{r}^k, \hat{P}^k) \geq V_1^{\pi^{\epsilon_k,*}}(\mu; r, P)$, for any $k \in [C'', K]$, and the lemma is proved. $\square$

The pivotal step is to leverage optimism of $\pi^k$ to further decompose the fourth term on the RHS of (20) and utilize the projected dual update to transfer it into the form of $\lambda^k(\lambda^k - \lambda^{k+1})$. The following lemmas provide high probability bounds for the remaining two terms in the RHS of (20):

**Lemma C.6** (Restatement of Lemma 5.9). *On the good event $\mathcal{E}$,*

$$\sum_{k=C''}^{K} \left( \hat{V}_1^{\pi^{\epsilon_k,*}}(\mu; \tilde{r}^k, \hat{P}^k) - \hat{V}_1^{\pi^k}(\mu; \tilde{r}^k, \hat{P}^k) \right) = \tilde{\mathcal{O}}\left( \frac{H}{\tau - c^0} \sqrt{K} \right).$$

*Proof of Lemma 5.9.* Taking advantage of the optimism of $\pi^k$ and the definition of the dual update, we have the following decomposition:

$$\sum_{k=C''}^{K} \left( \hat{V}_1^{\pi^{\epsilon_k,*}}(\mu; \tilde{r}^k, \hat{P}^k) - \hat{V}_1^{\pi^k}(\mu; \tilde{r}^k, \hat{P}^k) \right) = \underbrace{\sum_{k=C''}^{K} \left( \frac{\lambda^k}{\eta^k}(\hat{V}_1^{\pi^{\epsilon_k,*}}(\mu; \tilde{c}^k, \hat{P}^k) - \hat{V}_1^{\pi^k}(\mu; \tilde{c}^k, \hat{P}^k)) \right)}_{(I)}$$

$$+ \underbrace{\sum_{k=C''}^{K} \left( \hat{V}_1^{\pi^{\epsilon_k,*}}(\mu; \tilde{r}^k, \hat{P}^k) - \frac{\lambda^k}{\eta^k}\hat{V}_1^{\pi^{\epsilon_k,*}}(\mu; \tilde{c}^k, \hat{P}^k) \right) - \left( \hat{V}_1^{\pi^k}(\mu; \tilde{r}^k, \hat{P}^k) - \frac{\lambda^k}{\eta^k}\hat{V}_1^{\pi^k}(\mu; \tilde{c}^k, \hat{P}^k) \right)}_{(II)}$$

Owing to the optimism of $\pi^k$, we know (II) $\leq 0$. Term (I) can then be upper bounded as follows:

$$(I) \overset{(a)}{\leq} \sum_{k=C''}^{K} \frac{\lambda^k}{\eta^k} \left( \tau - \epsilon_k - \hat{V}_1^{\pi^k}(\mu; \tilde{c}^k, \hat{P}^k) \right)$$

$$\overset{(b)}{\leq} \sum_{k=C''}^{K} \frac{1}{\eta^k} \left( \lambda^k(\lambda^k - \lambda^{k+1}) + \tau^2 \right)$$

$$\overset{(c)}{=} \sum_{k=C''}^{K} \frac{1}{\eta^k}(\frac{1}{2}(\lambda^k)^2 - \frac{1}{2}(\lambda^{k+1})^2) + \sum_{k=C''}^{K} \frac{1}{2\eta^k}(\lambda^{k+1} - \lambda^k)^2 + \sum_{k=C''}^{K} \frac{1}{\eta^k}\tau^2$$

$$\overset{(d)}{\leq} \frac{(\lambda^{C''})^2}{2\eta^{C''}} + \sum_{k=C''}^{K} \frac{H^2}{2\eta^k} + \tau^2 \sum_{k=C''}^{K} \frac{1}{\eta^k}$$

$$\overset{(e)}{\leq} \frac{(\sum_{k=1}^{C''} \epsilon_k + C''(H - \tau))^2}{2\eta^{C''}} + \sum_{k=C''}^{K} \frac{H^2}{2\eta^k} + \tau^2 \sum_{k=C''}^{K} \frac{1}{\eta^k}.$$

(a) holds by applying Lemma C.1 for $\pi = \pi^{\epsilon_k,*}$ and $\hat{V}_1^{\pi^{\epsilon_k,*}}(\mu; c, P) \leq \tau - \epsilon_k$.

(b) holds as follows: if $\lambda^{k+1} > 0$, we know

$$\tau - \epsilon_k - \hat{V}_1^{\pi^k}(\mu; \tilde{c}^k, \hat{P}^k) = \lambda^k - \lambda^{k+1}.$$

If $\lambda^{k+1} = 0$, we have

$$\lambda^k \leq \tau - \epsilon_k - \hat{V}_1^{\pi^k}(\mu; \tilde{c}^k, \tilde{P}^k) < \tau,$$

which indicates that $\lambda^k(\tau - \epsilon_k - \hat{V}_1^{\pi^k}(\mu; \tilde{c}^k, \hat{P}^k)) \leq \tau^2$. Thus,

$$\lambda^k(\tau - \epsilon_k - \hat{V}_1^{\pi^k}(\mu; \tilde{c}^k, \hat{P}^k)) \leq \lambda^k(\lambda^k - \lambda^{k+1}) + \tau^2.$$

Then, (c) holds since $-\epsilon\Delta = \frac{1}{2}\epsilon^2 - \frac{1}{2}(\epsilon + \Delta)^2 + \frac{1}{2}\Delta^2$, where $\epsilon = \lambda^k, \Delta = \lambda^{k+1} - \lambda^k$, (d) holds since $\eta^k$ is monotonically increasing and $(\lambda^k - \lambda^{k+1})^2 \leq (\hat{V}_1^{\pi^k}(\mu; \tilde{c}^k, \hat{P}^k) + \epsilon_k - \tau)^2 \leq H^2$, and (e) holds according to (19). $\square$

**Lemma C.7** (Restatement of Lemma 5.10). *On the good event $\mathcal{E}$,*

$$\sum_{k=C''}^{K} \left( \hat{V}_1^{\pi^k}(\mu; \tilde{r}^k, \hat{P}^k) - V_1^{\pi^k}(\mu; r, P) \right) = \tilde{\mathcal{O}}(H^2 \sqrt{|\mathcal{S}|^3|\mathcal{A}|K} + H^4|\mathcal{S}|^3|\mathcal{A}|).$$

*Proof of Lemma 5.10.* For any $k, s, a, h$, we have

$$
\begin{aligned}
|(\tilde{r}_h^k - r_h)(s,a)| &= |\tilde{r}_h^k(s,a) - \hat{r}_h^k(s,a) + \hat{r}_h^k(s,a) - r_h(s,a)| \\
&\leq (1 + |\mathcal{S}|H)\beta_h^k(s,a) + |\hat{r}_h^k(s,a) - r_h(s,a)| \\
&\leq (1 + |\mathcal{S}|H)\beta_h^k(s,a) + \beta_h^k(s,a) = (2 + |\mathcal{S}|H)\beta_h^k(s,a).
\end{aligned}
$$

By Lemma D.4, we have

$$
\sum_{k=C''}^{K} \left( \hat{V}_1^{\pi^k}(\mu; \tilde{r}^k, \hat{P}^k) - V_1^{\pi^k}(\mu; r, P) \right) \leq \tilde{\mathcal{O}}(H^2\sqrt{|\mathcal{S}|^3|\mathcal{A}|K} + H^4|\mathcal{S}|^3|\mathcal{A}|).
$$

$\square$

## C.3  Proof of Theorem 4.1

**Theorem C.8** (Regret and constraint violation bounds for OptPess-PrimalDual (Restatement of Theorem 4.1))**.** *On the good event $\mathcal{E}$,*

$$
Reg^{\textbf{\textit{OPPD}}}(K; r) = \tilde{\mathcal{O}} \left( \frac{H^3}{\tau - c^0}\sqrt{|\mathcal{S}|^3|\mathcal{A}|K} + \frac{H^5|\mathcal{S}|^3|\mathcal{A}|}{(\tau - c^0)^2} \right),
$$

$$
Reg^{\textbf{\textit{OPPD}}}(K; c) = \mathcal{O} \left( C''(H - \tau) + H^2\sqrt{|\mathcal{S}|^3|\mathcal{A}|C''} \right) = \mathcal{O}(1),
$$

*where $C'' = \mathcal{O}(\frac{H^4|\mathcal{S}|^3|\mathcal{A}|}{(\tau-c^0)^2} \log \frac{H^4|\mathcal{S}|^3|\mathcal{A}|}{(\tau-c^0)^2\delta'})$ is a coefficient independent of $K$.*

*Proof of Theorem 4.1.* The bounded constraint analysis has been provided in the previous subsection. Applying Lemmas 5.7, 5.8, 5.9, and 5.10 yields the regret bound in Theorem 4.1. $\square$

## D  Other supporting lemmas

**Lemma D.1** (Hoeffding's inequality [15])**.** *For independent zero-mean $1/2$-sub-Gaussian random variables $X_1, X_2, \ldots, X_n$,*

$$
\mathbb{P} \left( \frac{1}{n}\sum_{i=1}^{n} X_i \geq \epsilon \right) \leq \exp\left(-n\epsilon^2\right),
$$

$$
\mathbb{P} \left( \frac{1}{n}\sum_{i=1}^{n} X_i \leq -\epsilon \right) \leq \exp\left(-n\epsilon^2\right).
$$

**Lemma D.2** (Value difference lemma, [8, Lemma E.15])**.**

$$
\begin{aligned}
&V_1^\pi(\mu; g', P') - V_1^\pi(\mu; g, P) \\
=&\mathbb{E}_{\mu, P, \pi} \left[ \sum_{h=1}^{H} \left( g'(S_h, A_h) - g(S_h, A_h) + \sum_{s'}(P_h' - P_h)(s'|S_h, A_h)V_{h+1}^\pi(s'; g', P') \right) \Big| \mathcal{F}_{k-1} \right] \\
=&\mathbb{E}_{\mu, P', \pi} \left[ \sum_{h=1}^{H} \left( g'(S_h, A_h) - g(S_h, A_h) + \sum_{s'}(P_h' - P_h)(s'|S_h, A_h)V_{h+1}^\pi(s'; g, P) \right) \Big| \mathcal{F}_{k-1} \right],
\end{aligned}
$$

*where $g = g' = r, c$.*

**Lemma D.3** ([2, Theorem 6.1(i)])**.** *Suppose the transition function is $P$. For any mixed policy $\tilde{\pi} = B_\gamma\pi^1 + (1 - B_\gamma)\pi^2$, where $B_\gamma$ is a Bernoulli distributed random variable with mean $\gamma$. Then there exists a Markov policy $\hat{\pi}$ that*

$$
V_h^{\hat{\pi}}(s; r, P) = V_h^{\tilde{\pi}}(s; r, P), \quad \forall r, s, h.
$$

**Lemma D.4.** *Let $\mathcal{G}_{1:K}$ be a sequence of events such that $\mathcal{G}_k \in \mathcal{F}_{k-1}$ for each $k \in [K]$. Suppose $|\tilde{g}^k - g| \leq \alpha\beta^k$, $\alpha \geq 1$. On the good event $\mathcal{E}$, for any $K' \leq K$,*

$$\sum_{k=1}^{K'} \mathbb{1}(\mathcal{G}_k)|V_1^{\pi^k}(\mu; \tilde{g}^k, \hat{P}^k) - V_1^{\pi^k}(\mu; g, P)| \leq (3\alpha + 3\sqrt{2}H\sqrt{|\mathcal{S}|})H\sqrt{|\mathcal{S}||\mathcal{A}|K'_{\mathcal{G}}Z}$$

$$+ \tilde{\mathcal{O}}\left(\alpha H^3|\mathcal{S}|^2|\mathcal{A}|\right),$$

*where $\mathcal{G}_{1:K} = (\Omega)_{1:K}$ or $\mathcal{G}_{1:k} = \{V_1^{\pi^0}(\mu; \underline{c}^k, \hat{P}^k) \geq \frac{\tau+c^0}{2}\}_{k\in[K]}$, and $K'_{\mathcal{G}} = \sum_{k=1}^{K'} \mathbb{1}(\mathcal{G}_k)$.*

*Proof of Lemma D.4.* It can be proved following steps similar to those in the proof of Lemma 32 in [11], but utilizing Lemma D.5 for taking account of the predictable event sequence $\mathcal{G}_{1:K}$, and Lemma D.6 for bounding the leading order explicitly. For completeness, we include the detailed proof here.

$$\sum_{k=1}^{K'} \mathbb{1}(\mathcal{G}_k)\left|V_1^{\pi^k}(\mu; \tilde{g}^k, \hat{P}^k) - V_1^{\pi^k}(\mu; g, P)\right|$$

$$= \sum_{k=1}^{K'}\sum_{h=1}^{H}\sum_{s,a} \mathbb{1}(\mathcal{G}_k)q^{\pi_k}(s,a,h)\left|\tilde{g}_h^k(s,a) - g_h(s,a) + \sum_{s'}(\hat{P}_h^k - P_h)(s'|s,a)V_{h+1}^{\pi_k}(s'; \tilde{g}^k, \hat{P}^k)\right|$$

$$\overset{(a)}{\leq} \sum_{k=1}^{K'}\sum_{h=1}^{H}\sum_{s,a} \mathbb{1}(\mathcal{G}_k)q^{\pi_k}(s,a,h)\left[\alpha\beta_h^k(s,a) + \sqrt{\sum_{s'}2P(s'|s,a)(\beta_h^k(s,a))^2}\sqrt{|\mathcal{S}|H^2} + \frac{ZH|\mathcal{S}|}{N_h^k(s,a)\vee 1}\right]$$

$$+ \sum_{k=1}^{K'}\sum_{h=1}^{H}\sum_{s,a} \mathbb{1}(\mathcal{G}_k)q^{\pi_k}(s,a,h)\left|\sum_{s'}(\hat{P}_h^k - P_h)(s'|s,a)\left(V_{h+1}^{\pi_k}(s'; \tilde{g}^k, \hat{P}^k) - V_{h+1}^{\pi_k}(s'; g, P)\right)\right|$$

$$= \sum_{k=1}^{K}\sum_{h=1}^{H}\sum_{s,a} \mathbb{1}(\mathcal{G}_k)q^{\pi_k}(s,a,h)\left[(\alpha + \sqrt{2|\mathcal{S}|}H)\sqrt{\frac{Z}{N_h^k(s,a)\vee 1}} + \frac{ZH|\mathcal{S}|}{N_h^k(s,a)\vee 1}\right]$$

$$+ \sum_{k=1}^{K'}\sum_{h=1}^{H}\sum_{s,a} \mathbb{1}(\mathcal{G}_k)q^{\pi_k}(s,a,h)\left|\sum_{s'}(\hat{P}_h^k - P_h)(s'|s,a)\left(V_{h+1}^{\pi_k}(s'; \tilde{g}^k, \hat{P}^k) - V_{h+1}^{\pi_k}(s'; g, P)\right)\right|.$$

Since

$$\sum_{k=1}^{K'}\sum_{h=1}^{H}\sum_{s,a} \mathbb{1}(\mathcal{G}_k)q^{\pi_k}(s,a,h)\left|\sum_{s'}(\hat{P}_h^k - P_h)(s'|s,a)\left(V_{h+1}^{\pi_k}(s'; \tilde{g}^k, \hat{P}^k) - V_{h+1}^{\pi_k}(s'; g, P)\right)\right|$$

$$\overset{(b)}{\leq} \sum_{k=1}^{K'}\sum_{h=1}^{H}\sum_{s,a} \mathbb{1}(\mathcal{G}_k)q^{\pi_k}(s,a,h)\sum_{s'}\frac{\sqrt{2P_h(s'|s,a)Z}}{\sqrt{N_h^k(s,a)\vee 1}}\left|V_{h+1}^{\pi_k}(s'; \tilde{g}^k, \hat{P}^k) - V_{h+1}^{\pi_k}(s'; g, P)\right|$$

$$+ \sum_{k=1}^{K'}\sum_{h=1}^{H}\sum_{s,a} \mathbb{1}(\mathcal{G}_k)q^{\pi_k}(s,a,h)\frac{(1+\alpha)|\mathcal{S}|HZ}{N_h^k(s,a)\vee 1}$$

$$\overset{(c)}{\leq} \tilde{\mathcal{O}}(\alpha H^2|\mathcal{S}|^2|\mathcal{A}|) + |\mathcal{S}|\sqrt{\alpha|\mathcal{A}|H^3}\sqrt{\sum_{k=1}^{K'}\mathbb{1}(\mathcal{G}_k)\left|V_1^{\pi^k}(\mu; \tilde{g}^k, \hat{P}^k) - V_1^{\pi^k}(\mu; g, P)\right|} + |\mathcal{S}|\sqrt{\alpha|\mathcal{A}|H^3}$$

$$\times \sqrt{\sum_{k=1}^{K'}\sum_{h=1}^{H}\sum_{s,a}\mathbb{1}(\mathcal{G}_k)q^{\pi_k}(s,a,h)\left|\sum_{s'}(\hat{P}_h^k - P_h)(s'|s,a)\left(V_{h+1}^{\pi_k}(s'; \tilde{g}^k, \hat{P}^k) - V_{h+1}^{\pi_k}(s'; g, P)\right)\right|}$$

$$\overset{(d)}{\leq} \tilde{\mathcal{O}}(\alpha H^3|\mathcal{S}|^2|\mathcal{A}|) + \frac{3}{2}|\mathcal{S}|H\sqrt{\alpha|\mathcal{A}|H}\sqrt{\sum_{k=1}^{K'}\mathbb{1}(\mathcal{G}_k)\left|V_1^{\pi^k}(\mu; \tilde{g}^k, \hat{P}^k) - V_1^{\pi^k}(\mu; g, P)\right|},$$

we have

$$\sum_{k=1}^{K'} \mathbb{1}(\mathcal{G}_k) \left| V_1^{\pi^k}(\mu; \tilde{g}^k, \hat{P}^k) - V_1^{\pi^k}(\mu; g, P) \right|$$

$$\overset{(e)}{\leq} (2\alpha + 2\sqrt{2}H\sqrt{|\mathcal{S}|})H\sqrt{|\mathcal{S}||\mathcal{A}|K_{\mathcal{G}}'Z} + \tilde{\mathcal{O}}\left(H|\mathcal{S}||\mathcal{A}|(\alpha + |\mathcal{S}|H)\right) + \tilde{\mathcal{O}}(\alpha H^3 |\mathcal{S}|^2|\mathcal{A}|)$$

$$+ \frac{3}{2}|\mathcal{S}|H\sqrt{\alpha|\mathcal{A}|H} \sqrt{\sum_{k=1}^{K'} \mathbb{1}(\mathcal{G}_k) \left| V_1^{\pi^k}(\mu; \tilde{g}^k, \hat{P}^k) - V_1^{\pi^k}(\mu; g, P) \right|}$$

$$\overset{(f)}{\leq} (3\alpha + 3\sqrt{2}H\sqrt{|\mathcal{S}|})H\sqrt{|\mathcal{S}||\mathcal{A}|K_{\mathcal{G}}'Z} + \tilde{\mathcal{O}}\left(\alpha H^3 |\mathcal{S}|^2|\mathcal{A}|\right).$$

(a) and (b) hold due to the triangle inequality, the Cauchy-Schawarz inequality, and since $|V_1^{\pi^k}(\mu; \tilde{g}^k, \hat{P}^k) - V_1^{\pi^k}(\mu; g, P)| \leq |V_1^{\pi^k}(\mu; \tilde{g}^k, \hat{P}^k) - V_1^{\pi^k}(\mu; g, \hat{P}^k)| + |V_1^{\pi^k}(\mu; g, \hat{P}^k) - V_1^{\pi^k}(\mu; g, P)| \leq (\alpha + 1)H$. (c) follows by steps similar to those in Lemma 32 in [12]. (d) and (f) hold due to Lemma D.6. (e) holds due to Lemma D.5. $\qquad\square$

**Lemma D.5** ([17, Lemma 10] with a predictable events sequence). *Given a sequence of events $\mathcal{G}_{1:K}$ that $\mathcal{G}_k \in \mathcal{F}_{k-1}$ for each $k \in [K]$. With probability at least $1 - \delta$, for any $K' \leq K$*

$$\sum_{k=1}^{K'} \sum_{h=1}^{H} \sum_{s,a} \frac{\mathbb{1}(\mathcal{G}_k)q^{\pi^k}(s, a, h)}{N_h^k(s, a) \vee 1} \leq 4H|\mathcal{S}||\mathcal{A}| + 2H|\mathcal{S}||\mathcal{A}| \ln K_{\mathcal{G}}' + 4 \ln \frac{2HK}{\delta},$$

$$\sum_{k=1}^{K'} \sum_{h=1}^{H} \sum_{s,a} \frac{\mathbb{1}(\mathcal{G}_k)q^{\pi^k}(s, a, h)}{\sqrt{N_h^k(s, a) \vee 1}} \leq 6H|\mathcal{S}||\mathcal{A}| + 2H\sqrt{|\mathcal{S}||\mathcal{A}|K_{\mathcal{G}}'} + 2H|\mathcal{S}||\mathcal{A}| \ln K_{\mathcal{G}}' + 5 \ln \frac{2HK}{\delta},$$

*where $K_{\mathcal{G}}' = \sum_{k=1}^{K'} \mathbb{1}(\mathcal{G}_k)$ and $q^{\pi^k}$ is the occupancy measure of policy $\pi^k$, i.e., $q^{\pi^k}(s, a, h) = \mathbb{E}_{\mu, P, \pi^k}[\mathbb{1}(S_h^k = s, A_h^k = a)|\mathcal{F}_{k-1}]$.*

*Proof of Lemma D.5.* This lemma can be proved following steps similar to those in the proof of Lemma 10 in [17]. For completeness, we include the detailed proof here.

Let $\mathbb{1}_k(s, a, h)$ be the indicator of whether the pair $(s, a)$ is visited at step $h$ in episode $k$, so that $\mathbb{E}_k[\mathbb{1}_k(s, a, h)] = q^{\pi^k}(s, a, h)$. We decompose the first quantity as

$$\sum_{k=1}^{K'} \sum_{h=1}^{H} \sum_{s,a} \frac{\mathbb{1}(\mathcal{G}_k)q^{\pi^k}(s, a, h)}{N_h^k(s, a) \vee 1} = \sum_{k=1}^{K'} \sum_{h=1}^{H} \sum_{s,a} \frac{\mathbb{1}(\mathcal{G}_k)\mathbb{1}_k(s, a, h)}{N_h^k(s, a) \vee 1}$$

$$+ \sum_{k=1}^{K'} \sum_{h=1}^{H} \sum_{s,a} \frac{\mathbb{1}(\mathcal{G}_k)(q^{\pi^k}(s, a, h) - \mathbb{1}_k(s, a, h))}{N_h^k(s, a) \vee 1}.$$

The first term can be bounded as

$$\sum_{k=1}^{K'} \sum_{h=1}^{H} \sum_{s,a} \frac{\mathbb{1}(\mathcal{G}_k)\mathbb{1}_k(s, a, h)}{N_h^k(s, a) \vee 1} \leq 2H|\mathcal{S}||\mathcal{A}| + H|\mathcal{S}||\mathcal{A}| \ln K_{\mathcal{G}}'.$$

To bound the second term, we apply Lemma 9 in [17] with $Y_k = \sum_{h=1}^{H} \sum_{s,a} \frac{\mathbb{1}(\mathcal{G}_k)(q^{\pi^k}(s,a,h) - \mathbb{1}_k(s,a,h))}{N_h^k(s,a) \vee 1}, \lambda = 1/2$, and the fact

$$\mathbb{E}_k[Y_k^2] \leq \mathbb{E}_k\left[\left(\sum_{h=1}^{H} \sum_{s,a} \frac{\mathbb{1}(\mathcal{G}_k)\mathbb{1}_k(s, a, h)}{N_h^k(s, a) \vee 1}\right)^2\right]$$

$$= \mathbb{E}_k\left[\sum_{h=1}^{H} \sum_{s,a} \frac{\mathbb{1}(\mathcal{G}_k)\mathbb{1}_k(s, a, h)}{(N_h^k(s, a))^2 \vee 1}\right]$$

$$\leq \sum_{h=1}^{H} \sum_{s,a} \frac{\mathbb{1}(\mathcal{G}_k)q^{\pi^k}(s, a, h)}{N_h^k(s, a) \vee 1},$$

which show that with probability at least $1 - \delta/(2HK)$,

$$\sum_{k=1}^{K'}\sum_{h=1}^{H}\sum_{s,a} \frac{\mathbb{1}(\mathcal{G}_k)(q^{\pi^k}(s,a,h) - \mathbb{1}_k(s,a,h))}{N_h^k(s,a) \vee 1} \leq \frac{1}{2}\sum_{k=1}^{K'}\sum_{h=1}^{H}\sum_{s,a} \frac{\mathbb{1}(\mathcal{G}_k)q^{\pi^k}(s,a,h)}{N_h^k(s,a) \vee 1} + 2\ln\frac{2HK}{\delta}.$$

Combining these two bounds, rearranging, and applying a union bound over $k$ prove the first part of the lemma.

Similarly, we decompose the second quantity as

$$\sum_{k=1}^{K'}\sum_{h=1}^{H}\sum_{s,a} \frac{\mathbb{1}(\mathcal{G}_k)q^{\pi^k}(s,a,h)}{\sqrt{N_h^k(s,a) \vee 1}} = \sum_{k=1}^{K'}\sum_{h=1}^{H}\sum_{s,a} \frac{\mathbb{1}(\mathcal{G}_k)\mathbb{1}_k(s,a,h)}{\sqrt{N_h^k(s,a) \vee 1}}$$

$$+ \sum_{k=1}^{K'}\sum_{h=1}^{H}\sum_{s,a} \frac{\mathbb{1}(\mathcal{G}_k)(q^{\pi^k}(s,a,h) - \mathbb{1}_k(s,a,h))}{\sqrt{N_h^k(s,a) \vee 1}}.$$

The first term is bounded as

$$\sum_{k=1}^{K'}\sum_{h=1}^{H}\sum_{s,a} \frac{\mathbb{1}(\mathcal{G}_k)\mathbb{1}_k(s,a,h)}{\sqrt{N_h^k(s,a) \vee 1}} \leq 2H|\mathcal{S}||\mathcal{A}| + 2H\sqrt{|\mathcal{S}||\mathcal{A}|K_\mathcal{G}'}.$$

To bound the second term, we again apply Lemma 9 in [17] with $Y_k = \sum_{h=1}^{H}\sum_{s,a} \frac{\mathbb{1}(\mathcal{G}_k)(q^{\pi^k}(s,a,h) - \mathbb{1}_k(s,a,h))}{\sqrt{N_h^k(s,a)\vee 1}} \leq 1, \lambda = 1$, and the fact

$$\mathbb{E}_k[Y_k^2] \leq \mathbb{E}_k\left[\left(\sum_{h=1}^{H}\sum_{s,a}\frac{\mathbb{1}(\mathcal{G}_k)\mathbb{1}_k(s,a,h)}{\sqrt{N_h^k(s,a)\vee 1}}\right)^2\right] = \sum_{h=1}^{H}\sum_{s,a}\frac{\mathbb{1}(\mathcal{G}_k)q^{\pi^k}(s,a,h)}{N_h^k(s,a)\vee 1},$$

which shows that with probability at least $1 - \delta/(2HK)$,

$$\sum_{k=1}^{K'}\sum_{h=1}^{H}\sum_{s,a}\frac{\mathbb{1}(\mathcal{G}_k)(q^{\pi^k}(s,a,h) - \mathbb{1}_k(s,a,h))}{\sqrt{N_h^k(s,a)\vee 1}} \leq \sum_{k=1}^{K'}\sum_{h=1}^{H}\sum_{s,a}\frac{\mathbb{1}(\mathcal{G}_k)q^{\pi^k}(s,a,h)}{N_h^k(s,a)\vee 1} + \ln\frac{2HK}{\delta}.$$

Combining the first part of the lemma and employing a union bound prove the second part of the lemma. $\qquad\square$

**Lemma D.6.** *Suppose* $0 \leq x \leq a + b\sqrt{x}$, *for some* $a, b > 0$,

$$x \leq \frac{3}{2}a + \frac{3}{2}b^2.$$

*Proof of Lemma D.6.* By solving the equation of $x_0 = a + b\sqrt{x_0}$, we know

$$x \leq x_0 = \frac{1}{2}(\sqrt{4ab^2 + b^4} + 2a + b^2)$$

$$\leq \sqrt{ab^2} + \frac{b^2}{2} + a + \frac{b^2}{2}$$

$$\leq \frac{3}{2}a + \frac{3}{2}b^2.$$

$\qquad\square$

**Lemma D.7** ([21, Lemma 11]). *Let* $S(k)$ *be a random process,* $\Phi(k)$ *be its Lyapunov function with* $\Phi(0) = \Phi_0$ *and* $\Delta(k) = \Phi(k+1) - \Phi(k)$ *be the Lyapunov drift. Given an increasing sequence* $\{\varphi_t\}, \rho$ *and* $\nu_{\max}$ *with* $0 < \rho \leq \nu_{\max}$, *if the expected drift* $\mathbb{E}[\Delta(k)|S(k) = s]$ *satisfies the following conditions:*
*(i) There exists constants* $\rho > 0$ *and* $\varphi_t > 0$ *s.t.* $\Delta(k) \leq -\rho$ *when* $\Phi(k) \geq \varphi_k$, *and*
*(ii)* $|\Phi(k+1) - \Phi(k)| \leq \nu_{\max}$ *holds with probability one;*
*then we have*

$$e^{\zeta\Phi(k)} \leq e^{\zeta\Phi_0} + \frac{2e^{\zeta(\nu_{\max} + \varphi_k)}}{\zeta\rho},$$

*where* $\zeta = \rho/(\nu_{\max}^2 + \nu_{\max}\rho/3)$.

# E   Extensions (unknown $c^0$ with prior knowledge of $\pi^0$)

It may be noticed that the OptPess-LP algorithm can guarantee zero violation with high probability if a coefficient $c^{0'}$ with $V_1^{\pi^0}(\mu; c, P) \le c^{0'} < \tau$ is used to replace $c^0$ in the inputs. Thus when the value $V_1^{\pi^0}(\mu; c, P)$ is not known, one may first estimate an appropriate upper bound of $V_1^{\pi^0}(\mu; c, P)$, and call OptPess-LP using this upper bound. The details of choosing the upper bound are as follows.

We can execute policy $\pi^0$ sequentially until an estimate of $c^0$ with enough precision is obtained. Denote $\hat{c}^0(k)$ as the mean estimate after executing policy $\pi^0$ $k$ times. We stop the estimation process after $K''$ executions, where

$$K'' := \min \left\{ k \ge 1 : \tau - \hat{c}^0(k) \ge 3\sqrt{\frac{1}{kH} \log \frac{2K}{\delta''}} \right\} \wedge K.$$

By Hoeffding's inequality, with probability at least $1 - \delta''$,

$$|\hat{c}^0(k) - c^0| \le \sqrt{\frac{1}{kH} \log \frac{2K}{\delta''}}.$$

Thus with probability at least $1 - \delta''$, for $k = K'' - 1$,

$$\tau - 4\sqrt{\frac{1}{kH} \log \frac{2K}{\delta''}} < \hat{c}^0(k) - \sqrt{\frac{1}{kH} \log \frac{2K}{\delta''}} \le c^0,$$

which implies

$$K'' = k + 1 \le 1 + \frac{16 \log(2K/\delta'')}{H(\tau - c^0)^2}.$$

Then $K'' < K$ can then be guaranteed if $\frac{16 \log(2K/\delta'')}{H(\tau - c^0)^2} < K - 1$. After $K''$ executions, with probability at least $1 - \delta''$,

$$
\begin{aligned}
c^{0'} =& \hat{c}^0(K'') + \sqrt{\frac{1}{K''H} \log \frac{2K}{\delta''}} \\
=& \frac{1}{2}\left( \hat{c}^0(K'') - \sqrt{\frac{1}{K''H} \log \frac{2K}{\delta''}} \right) + \frac{1}{2}\left( \hat{c}^0(K'') + 3\sqrt{\frac{1}{K''H} \log \frac{2K}{\delta''}} \right) \\
\le& \frac{1}{2}c^0 + \frac{1}{2}\tau = \frac{1}{2}(c^0 + \tau).
\end{aligned}
$$

We can apply the OptPess-LP algorithm with a pessimistic estimate $\left( \hat{c}^0(K'') + \sqrt{\frac{1}{K''H} \log \frac{2K}{\delta''}} \right)$ of $c^0$. It also maintains zero constraint violation with the same order of regret since $1/(\tau - c^{0'}) \le 1/(\tau - \frac{1}{2}(c^0 + \tau)) = 2/(\tau - c^0)$.