# OpenReview forum: "Learning Policies with Zero or Bounded Constraint Violation for Constrained MDPs"
_NeurIPS.cc/2021/Conference — NeurIPS 2021 Poster_

### Official Review · Reviewer_5BQY · 2021-07-15

**Rating:** 6
**Confidence:** 5

**Summary:**

This paper studies safety in reinforcement learning. It guarantees a zero constraint violation happens with high probability when strictly safe policy is known. The algorithm employs optimism in the face of uncertainty and achieves a regret of O(\sqrt{K}). For the case with no known strictly safe policy, it is proven that  the constraint violation is bounded.

**Limitations And Societal Impact:**

I believe this work does not have any potential negative societal impact.

**Main Review:**

The paper is well-written and the problem formulation and the contributions are clearly stated. However, numerical experiments that corroborate the theoretical guarantees of the paper are missing in the current version and I highly recommend to add some in the revision.

Questions:

1. My first question is whether the authors can extend their analysis to the case with focus on deterministic policy?
2. My second question is whether either of the regret bounds and the constraint violation bound are tight in terms of c^0?
3. Could the authors comment  on an example of safety-critical system where it is reasonable to assume the knowledge of a safe policy?



**Time Spent Reviewing:**

3 hours

---

> ### Author Response · Authors · 2021-08-10
> **Response to Reviewer 3**
>
> Thank you very much for your comments and suggestions. Please see our detailed response below. Please note that reference numbers and line numbers are based on the supplementary material, while additional references are marked in [R].
>
> 1. "... numerical experiments that corroborate the theoretical guarantees of the paper are missing in the current version ..."
>
> *Response:* As the reviewer observes, the main objective of our work is indeed to contribute to the fundamental theory of safe RL by developing learning algorithms that can provably guarantee no safety constraint violation (or bounded safety constraint violation) during learning. Following the tradition of important papers that have advanced our understanding of such problems in a tabular setting [R5, R4, R6, R1, 16, 11, 9], or that use linear function approximation [R7, R2, R3, 9], we have focused on developing the theoretical foundations, and have not included simulation results.
>
> However, after receiving the reviewers' comments, we have simulated our algorithms on a small tabular CMDP problem. We have also compared our results with two algorithms proposed in [11], namely the OptCMDP-bonus algorithm and the OptDual-CMDP algorithm. Since OpenReview does not allow us to include simulation plots in our response, we provide a qualitative description of the simulation results we have observed. As expected, our algorithms achieved zero or bounded constraint violation regret whereas the algorithms proposed in [11] incurred an $\tilde{O}(\sqrt{K})$ constraint violation regret. In particular, our OptPess-LP algorithm achieved zero constraint violation regret while incurring an $\tilde{O}(\sqrt{K})$ regret with respect to the objective. We verified the $\tilde{O}(\sqrt{K})$ behavior by dividing the cumulative regret by $1/\sqrt{K}$ and observing that this ratio converges to a constant. We have also observed that during the initial phase of learning, due to the forced exploration via the strictly safe policy $\pi^{0}$, the regret of the OptPess-LP algorithm increases linearly. However, this linear increase stops after a number of episodes and follows an $\tilde{O}(\sqrt{K})$ increase after that, as predicted by the theory. Due to this initial linear increase, the objective regret of our  OptPess-LP algorithm is slightly more than that of the algorithms proposed in [11]. The simulation of our OptPess-PrimalDual algorithm also showed results as predicted by the theory.
>
> 2. "... whether the authors can extend their analysis to the case with a focus on deterministic policy?"
>
> *Response:* The optimal policy of a CMDP is generally not deterministic. In general, it requires randomization to take full advantage of any allowed safety violation [2]. So, restricting the learning policies to the set of deterministic policies will generally lead to suboptimal outcomes.
>
> 3. "... whether either of the regret bounds and the constraint violation bound is tight in terms of $c^0$?"
>
> *Response:* The cost of the strictly safe policy $\pi^{0}$, denoted as $c^{0}$, appears in the regret as the multiplicative constant of the $\sqrt{K}$ term. In particular, it appears as $\frac{1}{(\tau - c^{0})}$. We are not sure if the optimal bound will be of the form $f(\frac{1}{(\tau - c^{0})})$, where $f(\cdot)$ is some function.
>
> 4. "... comment on an example of a safety-critical system where it is reasonable to assume the knowledge of a safe policy"
>
> *Response:* In general, in safety-critical systems, it is highly desirable to have knowledge of a safe policy. A well-known example is the dead man's stick (or dead man's switch), originally employed to stop trains. In modern cars with automatic lane-keeping and intelligent cruise control, if the driver does not have his/her hands on the steering wheel, the car stops after a warning. In the context of a nuclear reactor, the safe policy consists of dousing the rods or withdrawing them.
> (Chernobyl was a bad design since it required power to pump water to douse the reactor, and hence unsafe).
>
> Indeed the assumption of the knowledge of a strictly safe policy is reasonable in the context of a number of safety-critical systems such as autonomous vehicles, power systems, and communication networks. For example, consider the problem of designing an RL algorithm for an autonomous racing car. Here, the objective is to complete several loops around the racing track, with the safety constraint being to stay within the track. A control policy that drives the car at a very slow speed can always ensure that the safety constraint is satisfied. A safe learning algorithm can thus start from such a safe control policy and slowly increase the speed without violating the safety constraint.
>
> Similarly, in power systems, the existing control algorithms for voltage regulation are designed very conservatively to avoid any violation of the voltages beyond a specified value. In resilient communication networks, the coding and modulation schemes are selected conservatively to ensure a minimum data rate at all times. A safe RL algorithm can use such existing conservative algorithms as a known strictly safe policy.
>
> *References:*
>
> [R1] C. Jin, Z. Allen-Zhu, S. Bubeck, and M. I. Jordan. "Is q-learning provably efficient?" In Proceedings of the 32nd International Conference on Neural Information Processing Systems, pages 4868–4878, 2018.
>
> [R2] C. Jin, Z. Yang, Z. Wang, and M. I. Jordan. "Provably efficient reinforcement learning with linear function approximation." In Conference on Learning Theory, pages 2137–2143. PMLR, 2020.
>
> [R3] Q. Cai, Z. Yang, C. Jin, and Z. Wang. "Provably efficient exploration in policy optimization." In International Conference on Machine Learning, pages 1283–1294. PMLR, 2020.
>
> [R4] C. Dann and E. Brunskill. "Sample complexity of episodic fixed-horizon reinforcement learning." Advances in Neural Information Processing Systems, 28:2818–2826, 2015.
>
> [R5] T. Jaksch, R. Ortner, and P. Auer. "Near-optimal regret bounds for reinforcement learning." Journal of Machine Learning Research, 11(4), 2010.
>
> [R6] M. G. Azar, I. Osband, and R. Munos. "Minimax regret bounds for reinforcement learning." In Proceedings of the 34th International Conference on Machine Learning-Volume 70, pages263–272, 2017.
>
> [R7] L. Yang and M. Wang. "Reinforcement learning in feature space: Matrix bandit, kernels, and regret bound." In International Conference on Machine Learning, pages 10746–10756. PMLR, 2020.

---

### Official Review · Reviewer_cdW5 · 2021-07-16

**Rating:** 6
**Confidence:** 2

**Summary:**

This paper presents two novel reinforcement learning algorithms for CMDPS, one which has no constraint violations with high probability, but requires a known safe policy, and another which bounds the probability of constraint regret, but only requires the existence of a safe policy. The first algorithm picks pessimistically safe policies if they obtain a high enough value, but chooses a best reward policy otherwise. The second algorithm iteratively chooses the highest value policy subject to a dynamically tightening constraint. Both algorithms have a higher regret than previous approaches, but are provably safe  in the respective metrics

**Limitations And Societal Impact:**

No discussion on societal impact. Some limitations are mentioned.

**Main Review:**

Strengths:

Broadly speaking the paper is well written.

The knowledge of safe policies are reasonable assumptions, and the two distinct cases seem natural. The relevance and novelty is clear.

The description of both procedures are clear and reproducible.

The theoretical aspects of the paper are thorough.


Weaknesses:
The paper consistently refers to notation before it is introduced, making it difficult to read. While it is hard to summarise the work in the introduction without a formal problem definition, some of the more important terms, like regret, should be explained informally.

The problem formulation needs to more clearly state what parts of the CMDP are known vs unknown to the agent at start.

The two approaches lack intuition. With the appendix, it's clear why certain bounds were chosen, but motivation that ties to the problem would help the reader's understanding.

Finally, whilst I appreciate the focus of this paper is theory, it is presenting two algorithms, so I think an empirical evaluation would greatly strengthen the contribution. This could be done by moving some of the results in Section 5 to the appendix.


Specific comments/questions

* Is assumption 2.1 common? Further intuition on its need, and how strong it is, should be given

* In line 129, I don't follow the use of sup. Shouldn't it be lim k->infty?

* point (iii) in line 132 feels a bit irrelevant in the context of this paper

===UPDATE===
After reading the other reviews and the authors' thorough responses, I think the paper should probably be accepted, assuming some of the insights and clarifications provided in the responses are added to the paper.

**Time Spent Reviewing:**

3

---

> ### Author Response · Authors · 2021-08-10
> **Response to Reviewer 2**
>
> Thank you very much for your comments and suggestions. We are encouraged by the fact that the reviewer finds our paper "well written", "relevance and novelty are clear", and "theoretical aspects of the paper are thorough". Please see our response below with respect to the specific comments. Please note that reference numbers and line numbers are based on the supplementary material, while additional references are marked in [R].
>
> 1. *On the notations in the introduction*
>
> *Response:* As suggested by the reviewer, for added clarity, we will add a description below Table 1 on the meanings of all the notations. This will supplement the explanation of the notation $K$ in the abstract (Line 4), and other notations in Lines 57 to Line 59 as they first appear in the introduction.
>
> 2. *"... some of the more important terms, like regret, should be explained informally."*
>
> *Response:* Following the suggestion by the reviewer, we will add an intuitive explanation of "regret" in the revised manuscript. This will supplement the formal definition currently provided in Lines 123 - 129.
>
> 3. *"... needs to more clearly state what parts of the CMDP are known vs unknown ..."*
>
> *Response:*  In the problem formulation, $P, r, c$ are unknown (as stated in Line 43) and  $\tau, c^0$ are known (as stated in Lines 57 - 58). Similar to the standard setting in RL literature [R1, 16, 8], we have also assumed that  $|\mathcal{S}|, |\mathcal{A}|, H$ are known. As suggested by the reviewer, we will explicitly write these as a single remark in the revised manuscript.
>
> 4. *"With the appendix, it's clear why certain bounds were chosen, but the motivation that ties to the problem would help the reader's understanding."*
>
> *Response:* As suggested by the reviewer, we will add the following explanation to aid readers' understanding of the bounds that supplement the intuition provided in the introduction in Lines 41 - 50, and in Lines 134 - 138, and 161 - 164.
>
> The main objective considered in this paper is to learn about the unknown system so as to earn a high reward while incurring a reduced constraint-violation cost. The intuitive idea is to combine optimism about the reward with pessimism about the constraint violation cost, an approach called *Optimistic Pessimism in the Face of Uncertainty*.
>
> In the OptPess-LP algorithm (Algorithm 1), the optimistic aspect consists of inflating the reward so as to incentivize the algorithm to explore policies that can visit new state-action pairs, while the pessimistic aspect consists of inflating the constraint violation costs and thereby disincentivizing the algorithm from using exploration policies that can violate safety constraints. The upper bounds on rewards and constraint violation costs are chosen to guarantee zero or bounded safety constraint violation during learning while achieving an $\tilde{O}(\sqrt{K})$ regret with respect to the reward objective.
> Without knowledge of a strictly safe policy, it is impossible to guarantee zero constraint violation in unknown systems since the agent needs to learn from scratch. Combining these bounds, we define a "pessimistically safe policy set" $\Pi^k$, given in Line 147 and Eq. (9), for which any policy $\pi \in \Pi^k$ is safe, but yet contains policies that maximize the reward.
>
> The OptPess-PrimalDual algorithm (Algorithm 2) does not require knowledge of a strictly safe policy, but only requires knowledge of a feasible value of total constraint cost. The optimism in this algorithm consists of inflating the reward estimate as well as *reducing* the constraint-violation cost. The pessimism consists of strengthening the safety constraint by a pessimistic term $\epsilon_{k}$, which is gradually decreased as the algorithm learns enough information from the environment
> so as to learn an optimal policy.
>
> 5. *"Finally, whilst I appreciate the focus of this paper is theory, it is presenting two algorithms, so I think an empirical evaluation would greatly strengthen the contribution."*
>
> *Response:* As the reviewer observes, the main objective of our work is indeed to contribute to the fundamental theory of safe RL by developing learning algorithms that can provably guarantee no safety constraint violation (or bounded safety constraint violation) during learning. Following the tradition of important papers that have advanced our understanding of such problems in a tabular setting [R5, R4, R6, R1, 16, 11, 9], or that use linear function approximation [R7, R2, R3, 9], we have focused on developing the theoretical foundations, and have not included simulation results.
>
> However, after receiving the reviewers' comments, we have simulated our algorithms on a small tabular CMDP problem. We have also compared our results with two algorithms proposed in [11], namely the OptCMDP-bonus algorithm and the OptDual-CMDP algorithm. Since OpenReview does not allow us to include simulation plots in our response, we provide a qualitative description of the simulation results we have observed. As expected, our algorithms achieved zero or bounded constraint violation regret whereas the algorithms proposed in [11] incurred an $\tilde{O}(\sqrt{K})$ constraint violation regret. In particular, our OptPess-LP algorithm achieved zero constraint violation regret while incurring an $\tilde{O}(\sqrt{K})$ regret with respect to the objective. We verified the $\tilde{O}(\sqrt{K})$ behavior by dividing the cumulative regret by $1/\sqrt{K}$ and observing that this ratio converges to a constant. We have also observed that during the initial phase of learning, due to the forced exploration via the strictly safe policy $\pi^{0}$, the regret of the OptPess-LP algorithm increases linearly. However, this linear increase stops after a number of episodes and follows an $\tilde{O}(\sqrt{K})$ increase after that, as predicted by the theory. Due to this initial linear increase, the objective regret of our  OptPess-LP algorithm is slightly more than that of the algorithms proposed in [11]. The simulation of our OptPess-PrimalDual algorithm also showed results as predicted by the theory.
>
> 6. *Specific comments*
>
> * "Is assumption 2.1 common?"
>
> *Response:* The assumption of sub-Gaussian noise is common in the analysis of RL theory literature [8, R2, R3, 9]. It is weaker than assuming that the noise has bounded support.
>
> * "In line 129, I don't follow the use of sup. Shouldn't it be lim k->infty?"
>
> *Response:* Here we use "$\sup$" (denoting the supremum of a sequence), since the limit is not guaranteed to exist.
>
> * "point (iii) in Line 132 feels a bit irrelevant in the context"
>
> *Response:* The comment about "fairness" was included to address the larger issue of ethics in machine learning. However, since the reviewer suggests it is a bit tangential to the main result of the paper, we will remove this in the revised manuscript.
>
> *References:*
>
> [R1] C. Jin, Z. Allen-Zhu, S. Bubeck, and M. I. Jordan. "Is q-learning provably efficient?" In Proceedings of the 32nd International Conference on Neural Information Processing Systems, pages 4868–4878, 2018.
>
> [R2] C. Jin, Z. Yang, Z. Wang, and M. I. Jordan. "Provably efficient reinforcement learning with linear function approximation." In Conference on Learning Theory, pages 2137–2143. PMLR, 2020.
>
> [R3] Q. Cai, Z. Yang, C. Jin, and Z. Wang. "Provably efficient exploration in policy optimization." In International Conference on Machine Learning, pages 1283–1294. PMLR, 2020.
>
> [R4] C. Dann and E. Brunskill. "Sample complexity of episodic fixed-horizon reinforcement learning." Advances in Neural Information Processing Systems, 28:2818–2826, 2015.
>
> [R5] T. Jaksch, R. Ortner, and P. Auer. "Near-optimal regret bounds for reinforcement learning." Journal of Machine Learning Research, 11(4), 2010.
>
> [R6] M. G. Azar, I. Osband, and R. Munos. "Minimax regret bounds for reinforcement learning." In Proceedings of the 34th International Conference on Machine Learning-Volume 70, pages263–272, 2017.
>
> [R7] L. Yang and M. Wang. "Reinforcement learning in feature space: Matrix bandit, kernels, and regret bound." In International Conference on Machine Learning, pages 10746–10756. PMLR, 2020.

---

### Official Review · Reviewer_uBAh · 2021-07-16

**Rating:** 7
**Confidence:** 3

**Summary:**

The paper studies the problem of safety in learning, within the framework of tabular episodic Constrained MDPs (unknown transition, rewards and costs). The authors propose a method based on the principle of optimistic pessimism in face of uncertainty that achieves a regret of $\mathcal{O}(\sqrt{K})$ for the rewards ($K$ being the number of episodes) that is comparable to the existing methods. The main advantage of the proposed approach is that it achieves a smaller constraint violation: particularly zero constraint violation when an initial safe policy is given, and bounded constraint violation when no safe policy is known (but it is known that one such policy exists). The known policy case is based on the LP formulation of solving the pessimistic CMDP, and the unknown policy case uses a  primal-dual approach.

**Limitations And Societal Impact:**

yes

**Main Review:**

### Originality:

The proposed methods are inspired by safe exploration in constrained bandits but the extension and application to the constrained RL problem are novel and valuable. The authors do a good job of citing the related literature and differentiating their approach from the others.

### Quality:

The submission is technically sound with all the claims being supported sufficiently. All the assumptions and limitations are stated clearly.

### Clarity:

The paper is clear and easy to follow. Although I believe some aspects of the paper can be improved wrt clarity, particularly:
-  All the analysis depends on how the high probability good event is defined (Appendix A). I think the authors can do a better job in explaining the details in this section. For instance, when the author’s define $\mathcal{E}^{gen}$ the Hoeffding’s, Berstein and Union bound are all applied in a single step and it is not obvious how the $\frac{1}{4(1+|S|)}$ term shows up in the denominator for $\delta’$. I think it will be useful for the readers if the intermediate steps are also provided.
- The results depend heavily on Lemma D.5, the proof for which is referred to another related work. However, the referenced work uses different notation and setting that makes it difficult to draw the connections. I think it will be useful if the proofs for Lemma D.5 (and even D.4) can be presented in the work itself for the sake of completeness.
- Another point that can be highlighted a bit further is how the threshold for the first condition in Eq 9 ($\frac{\tau+c^0}{2}$) is derived. Some discussion on that will add to the clarity of the approach.



### Significance:

The paper is technically sound and novel with strong results which I believe will be useful for developing further safe/bounded-constraint algorithms in this setting.


**Time Spent Reviewing:**

4

---

> ### Author Response · Authors · 2021-08-10
> **Response to Reviewer 1**
>
> Thank you very much for your comments and suggestions to improve the clarity of the presentation/proofs. We are encouraged by the fact that the reviewer finds our paper "technically sound and novel with strong results" and "the paper is clear and easy to follow". Please see our response below with respect to the specific comments by the reviewer. Please note that reference numbers and line numbers are based on the supplementary material.
>
> 1. *"... it will be useful for the readers if the intermediate steps are also provided."*
>
> *Response (intermediate steps in the proof):* Define the event $\mathcal{E}^{gen} := \left((\cup_{n=1}^K F_n^r \cup F_n^c \cup F_n^P \cup \tilde{F}_n^P)^C  \cap \mathcal{E}_\Omega (\delta/4)  \cap \mathcal{E}_0 (\delta/4) \right).$ By the definition of $\mathcal{E}_\Omega (\delta/4)$ and $\mathcal{E}_0 (\delta/4)$ and Lemma D.5, event $\mathcal{E}_\Omega (\delta/4) \cap \mathcal{E}_0 (\delta/4)$ occurs with probability at least $1 - \delta/2$. So, to show that $\mathbb{P}(\mathcal{E}^{gen}) \geq 1 - \delta$, it is sufficient to show that $\mathbb{P}(\cup_n F_n^r \cup F_n^c \cup F_n^P \cup \tilde{F}_n^P) \leq \delta/2$.
>
> Recall the definition of the confidence intervals as stated in the proof of Lemma A.1: $\beta(n) := \sqrt{\frac{1}{n \lor 1}Z}$ and $\tilde{\beta}^k_h(s' | s, a) := \sqrt{\frac{2 P(s' | s, a)}{N^k_h(s, a) \lor 1} Z} + \frac{Z}{3 N^k_h(s, a) \lor 1}$, where $Z := \log(16|\mathcal{S}|^2 |\mathcal{A}|HK/\delta)$. Note that $\delta/16 \le \delta |\mathcal{S}|/4(1+|\mathcal{S}|) =: \delta'$. Now, it is straightforward to show the following:
> *  Using Hoeffding’s inequality, $ \mathbb{P}(\cup_{n=1}^K F_n^r) \leq |\mathcal{S}||\mathcal{A}|HK \frac{\delta}{16|\mathcal{S}|^2 |\mathcal{A}|HK} \leq \frac{\delta'}{|\mathcal{S}|}$.
> * Using Hoeffding’s inequality, $ \mathbb{P}(\cup_{n=1}^K F_n^c) \leq |\mathcal{S}||\mathcal{A}|HK \frac{\delta}{16|\mathcal{S}|^2 |\mathcal{A}|HK} \leq \frac{\delta'}{|\mathcal{S}|}$.
> * Using Hoeffding’s inequality, $\mathbb{P}(\cup_{n=1}^K F_n^P) \leq |\mathcal{S}|^2|\mathcal{A}|HK \frac{\delta}{16|\mathcal{S}|^2 |\mathcal{A}|HK} \leq \delta'$.
> * Using Bernstein's inequality, $\mathbb{P}(\cup_{n=1}^K \tilde{F}_n^P) \leq |\mathcal{S}|^2|\mathcal{A}|HK \frac{\delta}{16|\mathcal{S}|^2 |\mathcal{A}|HK} \leq \delta'$.
>
> Now, using union bound,
> $\mathbb{P}(\cup_{n=1}^K F_n^r \cup F_n^c \cup F_n^P \cup \tilde F_n^P) \le \mathbb{P}(\cup_{n=1}^K F_n^r) + \mathbb{P}(\cup_{n=1}^K F_n^c) + \mathbb{P}(\cup_{n=1}^K F_n^P) + \mathbb{P}(\cup_{n=1}^K \tilde{F}_n^P) \le (2 + 2 / |\mathcal{S}|) \delta' = \delta/2$,
> which completes the proof.
>
> 2. *"I think it will be useful if the proofs for Lemma D.5 (and even D.4) can be presented in the work itself for the sake of completeness."*
>
> *Response:* We had omitted the proof for conciseness, since, as mentioned in the manuscript, Lemma D.5 can be proved by following the same steps as in the proof of Lemma 10 in [16]. However, following the reviewer's comment, we will include the detailed proof of Lemma D.5 and Lemma D.4 in the revised manuscript.
>
> 3. *"Another point that can be highlighted a bit further is how the threshold for the first condition in Eq 9 (on $(\tau + c^0)/2$) is derived"*
>
> *Response:* As shown in Lemma 5.1, any policy in the set {$\pi: V_1^{\pi}(\mu; \underline{c}^k, \hat{P}^k) \leq \tau$} (if this set is non-empty) will not violate the constraint. However, because of our pessimistic approach, the set  {$\pi: V_1^{\pi}(\mu; \underline{c}^k, \hat{P}^k) \leq \tau$} may be empty in the initial phase of learning. So, we simply use the strictly safe policy $\pi^{0}$ until the above mentioned set becomes non-empty. The inequality $V_1^{\pi^0}(\mu; \underline{c}^k, \hat{P}^k) < (\tau + c^0)/2$ (c.f. Eq.(9)) provides a sufficient condition for the above mentioned set to become non-empty. We have rigorously shown this in the proof of Lemma 5.2. We will include this intuitive explanation in the revised manuscript.

---

### Decision · Program_Chairs · 2021-09-27

**Decision:**

Accept (Poster)

**Comment:**

This paper studies the problem of learning episodic CMDPs, which provides improved theoretical results comparing to previous works. This work is technically sound and novel with strong results. All the reviewers have reached a consensus on the acceptance of this work. We suggest that the authors further polish their paper to prepare a camera-ready version based on the reviewers’ comments.